# Global nonlinear approach for mapping parameters of neural mass models

**Dominic M. Dunstan**[1,2]\*, **Mark P. Richardson**[3], **Eugenio Abela**[3], **Ozgur E. Akman**[1†], **Marc Goodfellow**[1,2]\*

**1** Department of Mathematics & Statistics, University of Exeter, Exeter, United Kingdom, **2** Living Systems Institute, University of Exeter, Exeter, United Kingdom, **3** Department of Basic and Clinical Neuroscience, King's College London, London, United Kingdom

† Deceased.
\* d.dunstan@exeter.ac.uk (DMD); m.goodfellow@exeter.ac.uk (MG)

## Abstract

Neural mass models (NMMs) are important for helping us interpret observations of brain dynamics. They provide a means to understand data in terms of mechanisms such as synaptic interactions between excitatory and inhibitory neuronal populations. To interpret data using NMMs we need to quantitatively compare the output of NMMs with data, and thereby find parameter values for which the model can produce the observed dynamics. Mapping dynamics to NMM parameter values in this way has the potential to improve our understanding of the brain in health and disease.

Though abstract, NMMs still comprise of many parameters that are difficult to constrain *a priori*. This makes it challenging to explore the dynamics of NMMs and elucidate regions of parameter space in which their dynamics best approximate data. Existing approaches to overcome this challenge use a combination of linearising models, constraining the values they can take and exploring restricted subspaces by fixing the values of many parameters *a priori*. As such, we have little knowledge of the extent to which different regions of parameter space of NMMs can yield dynamics that approximate data, how nonlinearities in models can affect parameter mapping or how best to quantify similarities between model output and data. These issues need to be addressed in order to fully understand the potential and limitations of NMMs, and to aid the development of new models of brain dynamics in the future.

To begin to overcome these issues, we present a global nonlinear approach to recovering parameters of NMMs from data. We use global optimisation to explore all parameters of nonlinear NMMs simultaneously, in a minimally constrained way. We do this using multi-objective optimisation (multi-objective evolutionary algorithm, MOEA) so that multiple data features can be quantified. In particular, we use the weighted horizontal visibility graph (wHVG), which is a flexible framework for quantifying different aspects of time series, by converting them into networks.

We study EEG alpha activity recorded during the eyes closed resting state from 20 healthy individuals and demonstrate that the MOEA performs favourably compared to single objective approaches. The addition of the wHVG objective allows us to better constrain the model output, which leads to the recovered parameter values being restricted to smaller regions of parameter space, thus improving the practical identifiability of the model. We then

**Data Availability Statement:** All code used in this study is given in a toolbox and is made publicly available and maintained as a GitHub repository (https://github.com/domdunstan/NerualMassModellingToolbox). Processed data are

publicly available on https://osf.io/f2vya/. Raw EEG data can be accessed via text files within the GitHub repository.

**Funding:** DMD acknowledges the support of an Engineering and Physical Sciences Research Council DTP studentship (ref 2407565). MPR was funded by the NIHR Biomedical Research Centre at the South London and Maudsley NHS Foundation Trust, and MRC Centre for Neurodevelopmental Disorders (ref MR/N026063/1). EA was funded by a Marie Sklodowska-Curie Actions Postdoctoral Fellowship from the European Commission (ref 750884). The funders had no role in study design, data collection and analysis, decision to publish, or preparation of the manuscript.

**Competing interests:** The authors have declared that no competing interests exist.

use the MOEA to study differences in the alpha rhythm observed in EEG recorded from 20 people with epilepsy. We find that a small number of parameters can explain this difference and that, counterintuitively, the mean excitatory synaptic gain parameter is reduced in people with epilepsy compared to control. In addition, we propose that the MOEA could be used to mine for the presence of pathological rhythms, and demonstrate the application of this to epileptiform spike-wave discharges.

## Author summary

EEG is a useful tool to study large scale brain activity. Mathematical models have been developed to help improve the understanding of the generation of signals recorded from the EEG during different brain states. The dynamics of these models are dependent on their inputs (or parameters) and hence it is important to explore the parameter combinations that result in model dynamics that approximate data. This allows us to better understand how the data were generated. However, due to the relative complexity of these models, finding the parameter combinations that explain data can be a cumbersome task and hence many studies make simplifications about how model and data are compared. In this study, we introduce methods that do not require these simplifying assumptions. Using these methods we demonstrate that different choices in the way we compare models and data can lead to differences in what we infer about the underlying mechanisms. However, we find that combining different choices into the same algorithm allows us to better approximate features of the data and better constrain model parameters. We apply our method to try to understand differences observed in the resting EEG between patients with epilepsy and controls. We find that the model explains these differences predominately by a reduced excitatory synaptic gain in patients with epilepsy. We also demonstrate the potential of this method to "mine" for different kinds of dynamics in high dimensional models.

## Introduction

Mathematical models are crucial for understanding the complex dynamics that emerge in the brain. Since the majority of recordings of human brain dynamics are derived from the macroscopic scale (i.e. generated in relatively large regions of tissue), macroscopic models that describe these dynamics have been extensively studied [1]. They are derived from a mixture of empirical observations [2] and simplifying assumptions about how the dynamics of large populations (ensembles, or masses) of neurons emerge [3]. They are popular as they provide a useful balance between being able to generate a number of features of electrographic recordings in health and disease [4–6] and their comparative simplicity [1]. They also offer the potential to understand large-scale dynamics in terms of physiological mechanisms [7–9], though more work is required to link "macroscopic mechanisms" to our understanding of the brain at the cellular and cellular network level [10].

Studies using macroscopic models vary depending upon whether and how they incorporate data. On the one hand, model parameters can be varied and changes in model dynamics quantified, for example, using parameter sweeps or numerical continuation [4, 9, 11, 12]. Model output under different parameter settings can then be compared qualitatively with aspects of the dynamics of real data *post hoc*. This kind of approach has proven useful in understanding

principles underlying the generation of healthy and pathological brain dynamics [4, 5, 13]. At the other extreme are methods that aim to link model output directly to data. This approach of "inverting" or "calibrating" the model from data allows us to describe (or quantify) the data directly in terms of the parameters of the model. Dynamic causal modelling (DCM) is a well-known example of this kind of approach, having at its core the use of mathematical models to interpret recordings of brain dynamics [14]. In all of these approaches, the underpinning idea is that the dynamics of the model change as parameter values are varied, and we try to find parameter values for which model output closely resembles or "fits" the data. In most studies of ongoing brain dynamics, and some studies of evoked responses, it is the frequency spectrum of the data and model that are compared. However, some studies, particularly those aiming to understand nonlinear brain dynamics in disorders like epilepsy, have incorporated other properties of signals, such as the distribution of amplitudes [8, 15].

Although macroscopic brain models provide a simplified, or abstract view of the mechanisms of the brain, nevertheless they comprise of many parameters, the values of which can be difficult to constrain. A classic example is the widely used Jansen and Rit neural mass model (NMM), which treats neuronal populations as input to output converters; afferent input causes a linear impulse response in the population membrane potential, which is converted nonlinearly into an outgoing firing density [6]. The mathematical equations chosen to model these mechanisms have 9 free parameters in their simplest form. The Liley model [16], which extends the Jansen and Rit model to include synaptic and membrane reversal potentials, has 22 parameters in its simplest form. These parameters, when varied, give NMMs the flexibility to produce different kinds of rhythms and evoked responses; i.e. at different locations in parameter space the model can produce different dynamics. Mathematically, this is due to changes in the vector field as parameters are varied. In nonlinear NMMs, this inevitably includes the presence of bifurcations, which can result in nearby parameter values giving rise to qualitatively different dynamics (see e.g. [13]).

It is extremely challenging to characterise the variability in dynamics systematically in more than a few dimensions, and certainly in the >8 dimensions which is typically the case for NMMs. Therefore, existing approaches for recovering the values of parameters of NMMs from data make simplifying assumptions. These include fixing or constraining the values of many parameters *a priori*, therefore essentially working in a lower dimensional space [15, 17, 18], and linearising the output of the model so that it can be evaluated quickly, facilitating a rapid exploration of parameter space [19, 20]. However, it is important to consider retaining the nonlinearities in NMMs for several reasons [21], which include theories for resting brain dynamics assuming proximity to bifurcations [22]; the potential importance of nonlinear mechanisms for producing dynamics of common brain rhythms (e.g. the alpha rhythm [23]) and the requirement of nonlinear mechanisms to produce pathological dynamics like those seen in epilepsy [12].

Exploring constrained regions of parameter space means that any inferences made from the model are only valid in the reduced regions of parameter space explored, i.e. naturally, results are only valid in the context of the prior assumptions made. Unless there is a high degree of confidence in the prior constraints, it is important to check whether different conclusions could have been drawn if we had worked in other plausible regions of parameter space. However, prior assumptions on the values of NMM parameters are not certain: their values can be difficult to constrain given our lack of understanding of how macroscopic brain dynamics are generated. Parameters of NMMs are often assumed to be analogous to parameters at the microscopic scale, where we have more knowledge. Examples are the time scales of synaptic responses, where commonly used parameter values have arisen from values identified at the microscopic scale [8]. In reality, the "corresponding" impulse responses in NMMs are

abstractions, and are produced in tissue containing many different cells communicating by many different mechanisms (as highlighted in, for example, [2, 8, 16]). We might therefore expect the parameters of neural mass postsynaptic potentials (PSPs) to have different values to the parameters of neuronal PSPs.

Fortunately, tools are available to search large, complex parameter spaces for model dynamics that match data. In particular, nondeterministic search heuristics such as evolutionary algorithms (EA) or particle swarm have been used previously to recover parameters of NMMs from data [15, 20, 24]. These algorithms incorporate objective functions that quantify the difference between simulations and data, and search parameter space to find parameter values that minimise this difference. Crucially, methods like EAs do not solely rely on local gradients; rather they incorporate mechanisms by which new parameter sets can be generated in disparate regions of parameter space. This is important for nonlinear models with bifurcations, where information about where is best to explore in parameter space may not necessarily be available locally. In addition, these methods are not based inherently on propagating probability distributions, so simplifying assumptions on the shape of these distributions, such as Gaussianity, which is unlikely in nonlinear models, do not have to be made. Here we focus on EAs, which have shown promise in parameter mapping for NMMs [15, 18, 25].

Central to the use and performance of EAs (and other parameter inference methods like DCM) is the definition of objective functions, which define how we measure the difference between model output and data. The choice of objective function can vary depending on the features of the data that we deem to be most important. It is common to quantify differences in frequency spectra [4, 20], but more nuanced properties of the data, for example, amplitude distributions or the presence of pre-defined features can also be used [15, 18]. This choice determines where local or global minima are positioned in parameter space, and this dictates which parameter values are deemed to yield model output that resembles the data. It is therefore important to begin to investigate the effect that different choices of objective functions have. When multiple data features are of interest, composite objective functions can be formed [15], though this can lead to local minima in which only one aspect of the objective is minimised. An alternative is to use multi-objective optimisation, in which the goodness of fit of model output is assessed on each objective individually [18, 26, 27]. This approach can better ensure that trade-offs between the different objectives are explored, while also allowing for a rank-based objective sorting during the optimisation. This is crucial when one does not have any *a priori* knowledge of how the objectives interact.

To begin to advance the tools that are available to recover parameters of NMMs and to address the above issues, herein we present a global optimisation approach that allows the exploration of all parameters of nonlinear NMMs simultaneously, in a minimally constrained way. We use multi-objective optimisation (multi-objective evolutionary algorithm, MOEA) so that features beyond the power spectrum can be incorporated. For the latter, we use the weighted horizontal visibility graph (wHVG), which incorporates aspects of the amplitude distribution (see Fig 1 herein), but has also been shown to be capable of distinguishing between different kinds of noise [28] and subtle features of nonlinear brain rhythms [29]. The (w)HVG transforms a time series into a network, where each time point is mapped to a node, and two time points (nodes) are connected by (weighted) edges only if a straight line connecting the amplitudes at those time points does not pass through the time series at an intermediate point (see Fig 1). In addition to the favourable properties mentioned above, this opens up the possibility of measuring the dynamics of time series in different ways within the same framework, using different graph theoretical measures [30]. Here we focus on the distribution of node degrees, which is the simplest such measure.

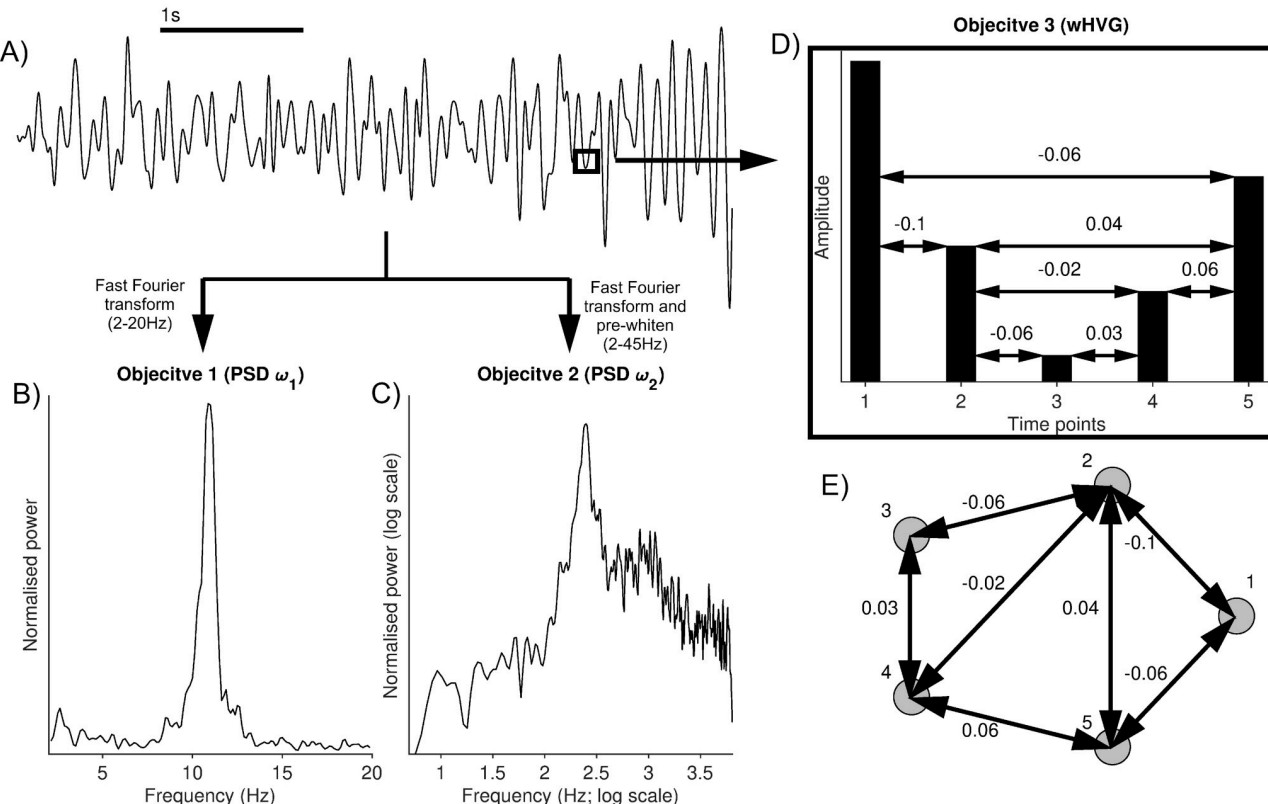

**Fig 1. Illustration of the objective calculations used to compare model output to data.** A) shows an exemplar z-scored EEG 5*s* time series (note that the actual data used in this work are 20*s* segments). A fast Fourier transform is applied to convert the time series into the frequency domain. B) shows the normalised power in the 2–20Hz range. This is used to calculate the objective for the SOEA20 and the first objective for the MOEA20. C) shows the normalised power after pre-whitening (see Methods). This is used to calculate the objective for the SOEA45 and the first objective for the MOEA45 approaches. D) shows the amplitude of 5 consecutive time points from the data in A), plotted as a bar chart. These time points have a resolution of 3.9*ms*. The conversion of the time series into the wHVG distribution for this section of the time series is shown (note that the whole time series is used to calculate the wHVG distribution in practice). The resulting network from this transformation is given in E). The sum of node weights of the wHVG is then used in the calculation of the second objective for the MOEA20 and MOEA45 approaches.

In order to test this approach, we use the MOEA to recover parameters of a NMM from resting EEG recorded from 20 healthy individuals. We demonstrate that the MOEA performs favourably compared to single objective evolutionary algorithms (SOEAs) in terms of finding model parameters for which the model well approximates the data. We also show that in using the wHVG as an objective, the MOEA is able to constrain other features of the data, namely the relative amplitude distribution and Hurst exponent. By mapping plausible NMM parameters in a global way, we demonstrate that seemingly subtle differences in the way in which model and data are compared can lead to substantial differences in the values of parameters that are recovered.

To demonstrate a potential use of the algorithm, we further apply it to resting EEG recorded from 20 people with epilepsy. We find that the resulting parameter distributions are largely the same as control, except for a small number of parameters that display differences. The largest difference found was for the mean excitatory synaptic gain parameter. Counterintuitively, we find this to be reduced in people with epilepsy, which hints at a potential protective mechanism at play during non-seizure epochs, or an effect of antiepileptic medication. It also provides evidence of the ability to recover mechanistic differences in the brain of people with epilepsy, from recordings that do not show seizure rhythms. Finally, we demonstrate the use of

this approach to uncover the existence of pathological dynamics in NMMs. We apply our algorithm to data comprising a classic epileptiform rhythm, the spike-wave discharge (SWD), and demonstrate that the Liley model can approximate the dynamics of this rhythm.

In summary, we present a global and nonlinear method for recovering NMM parameters from data. We demonstrate its ability to uncover differences in mechanisms of different brain states and to mine NMMs for the presence of different kinds of dynamics. Our results highlight the need to carefully consider how models and data are compared, as even subtly different approaches can lead to different inference.

## Methods

### Liley neural mass model

To model EEG, we apply the local spatially homogeneous version of the mean-field model first described in [31]. For a detailed breakdown of the model derivation, see [16, 32]. In short, the model describes the dynamics of a cortical macrocolumn in terms of the interaction between distinct populations of excitatory and inhibitory neurons. Each population is considered to be connected to both itself and the other population, with the assumption of fast-acting synapses. Subsequently, for $t \in [0, T]$ with $T \in \mathbb{R}_{>0}$, the model is comprised of the following set of coupled first and second-order differential equations:

$$\tau_e \frac{dh_e(t)}{dt} = h_e^{rest} - h_e(t) + \frac{h_e^{eq} - h_e(t)}{\mid h_e^{eq} - h_e^{rest} \mid} I_{ee}(t) + \frac{h_i^{eq} - h_e(t)}{\mid h_i^{eq} - h_e^{rest} \mid} I_{ie}(t),  \tag{1}$$

$$\tau_i \frac{dh_i(t)}{dt} = h_i^{rest} - h_i(t) + \frac{h_e^{eq} - h_i(t)}{\mid h_e^{eq} - h_i^{rest} \mid} I_{ei}(t) + \frac{h_i^{eq} - h_i(t)}{\mid h_i^{eq} - h_i^{rest} \mid} I_{ii}(t),  \tag{2}$$

$$\frac{d^2 I_{ee}(t)}{dt^2} + 2\gamma_{ee}\frac{dI_{ee}(t)}{dt} + \gamma_{ee}^2 I_{ee}(t) = \Gamma_e \gamma_{ee} \exp(1)(N_{ee}^\beta S_e(h_e(t)) + p(t)),  \tag{3}$$

$$\frac{d^2 I_{ei}(t)}{dt^2} + 2\gamma_{ei}\frac{dI_{ei}(t)}{dt} + \gamma_{ei}^2 I_{ei}(t) = \Gamma_e \gamma_{ei} \exp(1)(N_{ei}^\beta S_e(h_e(t)) + p_{ei}),  \tag{4}$$

$$\frac{d^2 I_{ie}(t)}{dt^2} + 2\gamma_{ie}\frac{dI_{ie}(t)}{dt} + \gamma_{ie}^2 I_{ie}(t) = \Gamma_i \gamma_{ie} \exp(1)(N_{ie}^\beta S_i(h_i(t))),  \tag{5}$$

$$\frac{d^2 I_{ii}(t)}{dt^2} + 2\gamma_{ii}\frac{dI_{ii}(t)}{dt} + \gamma_{ii}^2 I_{ii}(t) = \Gamma_i \gamma_{ii} \exp(1)(N_{ii}^\beta S_i(h_i(t))),  \tag{6}$$

where $S(\cdot)$ is a nonlinear sigmoid function that provides a transformation of the mean postsynaptic membrane potential into a mean firing rate. This mapping is defined as,

$$S_j(h_j(t)) = \frac{S_j^{max}}{\left(1 + exp(-\sqrt{2}\frac{(h_j(t) - \mu_j)}{\sigma_j})\right)},  \tag{7}$$

for subscripts $j = e, i$ representing the excitatory and inhibitory populations, respectively. Eqs 1 and 2 describe the temporal evolution of the mean soma membrane potentials. Furthermore, Eqs 3–6 describe the dynamics of the synaptically induced currents in response to afferent firing. For notation $I_{lk}$, the subscripts $lk$ correspond to synapses of type $l$ acting on neurons of type $k$. The terms $(h_l^{eq} - h_k(t))/(h_l^{eq} - h_k^{rest})$, for $l = e, i$ and $k = e, i$, incorporate that the mean

**Table 1. Parameter values with physiological interpretation, a typical chosen value and typically used bounds [16, 34].**

| Parameter | Interpretation | Typical Value | Range |
|---|---|---|---|
| $h_e^{rest}, h_i^{rest}$ | Mean excitatory/inhibitory membrane potential at rest | $-70mV, -70mV$ | $[-80mV, -60mV], [-80mV, -60mV]$ |
| $N_{ee}^\beta, N_{ei}^\beta, N_{ie}^\beta, N_{ii}^\beta$ | Number of excitatory—excitatory/excitatory—inhibitory/inhibitory—excitatory/inhibitory—inhibitory synaptic connections | 4000, 3034, 536, 536 | $[2000, 5000], [2000, 5000], [100, 1000], [100, 1000]$ |
| $\Gamma_e, \Gamma_i$ | Excitatory/inhibitory mean synaptic gain | $0.4mV, 0.8mV$ | $[0.1mV, 2mV], [0.1mV, 2mV]$ |
| $\gamma_e, \gamma_i$ | Excitatory/inhibitory postsynaptic potential rate constant | $0.3/ms, 0.065/ms$ | $[0.1/ms, 1/ms], [0.01/ms, 0.5/ms]$ |
| $\tau_e, \tau_i$ | Passive excitatory/inhibitory membrane time constant | $10ms, 10ms$ | $[5ms, 150ms], [5ms, 150ms]$ |
| $S_e^{max}, S_i^{max}$ | Maximum mean firing rate of excitatory/inhibitory population | $0.5/ms, 0.5/ms$ | $[0.05/ms, 0.5/ms], [0.05/ms, 0.5/ms]$ |
| $\mu_e, \mu_i$ | Excitatory/inhibitory firing rate thresholds | $-50mV, -50mV$ | $[-55mV, -40mV], [-55mV, -40mV]$ |
| $\sigma_e, \sigma_i$ | Excitatory/inhibitory firing threshold standard deviation | $5mV, 5mV$ | $[2mV, 7mV], [2mV, 7mV]$ |
| $h_e^{eq}, h_i^{eq}$ | Excitatory/inhibitory mean reversal potential | - | $[-20mV, -10mV], [-90mV, -65mV]$ |
| $p_{ee}, p_{ei}$ | Rate of excitatory-excitatory/excitatory-inhibitory input | - | $[0/ms, 10/ms], [0/ms, 10/ms]$ |
| $\xi$ | Standard deviation of noise perturbation | - | $[0, 10]$ |

magnitude of postsynaptic current flow, in response to synaptic activity, will be dependent on the mean soma membrane potential [16]. The function $p : [0, T] \to \mathbb{R}$ is modelled by a stochastic Gaussian process with a mean and variance given by parameters $p_{ee}$ and $\xi$, respectively, which accounts for extrinsic inputs to the neural mass (i.e. inputs from other unmodelled brain regions).

As established previously (for example, see [16, 31]), the mean soma membrane potential of the excitatory population ($h_e(t)$) is then taken to be the model output and is assumed to be linearly proportional to the EEG. Additionally, as in [20, 33], we assume $\gamma_{ee} = \gamma_{ei} = \gamma_e$ and $\gamma_{ie} = \gamma_{ii} = \gamma_i$. Finally, all other parameters are detailed in Table 1, along with physiological interpretations, typically chosen values and typically used bounds [16, 24].

### EEG data

Three data sets were used in total for this work. The first and the second were subsets of data from a previous study [35], and each consist of $20s$ epochs of EEG alpha activity recorded during the eyes closed resting state. The first set consisted of 20 healthy adults and the second set consisted of 20 adults diagnosed with focal epilepsy (FE). In the cohort that these were taken from, 56% were left lateralised. Further information can be found in [35] and at https://osf.io/f2vya/. All of these EEG recordings were taken on a NicoletOne system at 256Hz from 19 channels positioned according to the international 10–20 system, with two reference electrodes attached to the ear lobes. Data were re-referenced to the common average. For the purpose of studying the alpha rhythm, we restrict our analyses to the mean of the occipital electrodes. Furthermore, a 2Hz high-pass `Butterworth filter` was applied to the data, in order to remove low frequency components. The third data set used was a single scalp EEG recording taken from a patient with absence epilepsy at Inselspital Bern, Switzerland. This recording exhibited a clear SWD pattern. A 2Hz high-pass `Butterworth filter` was also applied to this data to remove low frequency components. Additionally, all data sets were z-score normalised.

### Model simulation

In order to numerically solve the model, Eqs 1–6 were converted into a set of ten first-order stochastic differential equations (SDEs). The Euler–Maruyama method was then used to numerically solve this set of SDEs, with zero initial conditions. A time step of 0.0125ms was

used to ensure accurate convergence. Each model output was calculated to match the EEG epoch length (20s for resting data, 2.3s for SWD data). To account for transient dynamics, a further 5s was calculated at the start of the model simulation and then removed. Note that we compare the model output $h_e(t)$ directly to the EEG because in this study we are neglecting contributions from other sources and therefore do not use a forward model. In future work with multiple sources, an appropriate forward model will need to be used.

The model was run using MATLAB's MEX interface [36]. This enables the implementation of C functions directly from MATLAB, providing enhanced computational efficiency compared to MATLAB functions alone. The authors would like to acknowledge the use of the University of Exeter's high-performance computing facility in carrying out this work.

## Multi-objective optimisation

Multi-objective optimisation aims to minimise $\mathbf{F}(\mathbf{x}) = (f_1(\mathbf{x}), f_2(\mathbf{x}), \ldots, f_d(\mathbf{x}))$, given a set of constraints on the inputs $\mathbf{x} = (x_1, x_2, \ldots, x_g)$. $\mathbf{F}(\mathbf{x})$ forms the $d$-dimensional objective space and $\mathbf{x}$ forms the $g$-dimensional decision space. Here, the decision space is the parameter space bounded within the ranges specified in Table 1. In the case $d = 1$, the problem becomes that of a single objective optimisation and hence an optimal solution can be defined as one that minimises the objective, given the constraints. For $d > 1$ the concept of optimality is more ambiguous, since solutions can exist whereby performance in one objective cannot be improved without hindering the performance of another objective. Such solutions are said to be Pareto optimal. Subsequently, we use an optimiser that employs the common approach of generating a set of points that satisfy the above condition, known as the non-dominated set of points. A decision vector $\mathbf{x}$ strictly dominates another decision vector $\mathbf{y}$ iff,

$$f_a(\mathbf{x}) \leq f_a(\mathbf{y}) \quad \forall a = 1, \ldots, d \quad \text{and}$$

$$f_a(\mathbf{x}) < f_a(\mathbf{y}) \quad \text{for} \quad \text{some} \quad a.$$

A genetic algorithm (GA) is a commonly used evolutionary global search heuristic which aims to determine a set of parameters that, upon simulation of the model, provide an output that recreates certain characteristics (or objectives) of the data. In this study, we utilise MATLAB's global optimisation toolbox to fit the model to the EEG data, by varying all the model parameters. Specifically, we use the `gamultiobj` function, which implements the `Non-dominated Sorting Genetic Algorithm-II` (NSGA-II) [37]. A central reason for using `NSGA-II` is because of the complex bifurcation structure that is known to exist in a NMM [38]. These bifurcations cause sudden changes in the objectives upon small parameter perturbations, which would render the ability of gradient-based optimisers to explore decision space, inadequate.

To formulate a GA, one must construct a fitness function based on objectives that the model is aimed to recreate, compared to the data. The GA then goes through a series of iterations (or generations) for which the model is simulated for an ensemble of parameter sets (the population). In this study, the initial population is obtained through a random Latin hypercube sampling of the parameters within their given physiological bounds. At each generation, every individual in the population is assigned a fitness score, based on how well that individual recreates the preassigned objectives. Each individual is then given a rank obtained from the fitness scores. Based on these ranks, the algorithm subsequently uses a tournament selection process, along with a crossover and mutation function, to create the population for the subsequent generation [37]. We use MATLAB's default values for the mutation and selection

function. However, we apply the scattered crossover function (as opposed to the default intermediate crossover), as we found this allowed for a better exploration of the feasible solutions in parameter space. As model simulations are stochastic, we simulate the model and calculate the objective function five times, and then assign the mean value obtained from these repeats as the objective for that parameter set.

To compare model output of the MOEA and SOEA approaches, we selected a single point from the non-dominated set obtained from the MOEAs. This point was chosen as the point that maps to the smallest Euclidean distance in objective space from the origin, after normalisation of the objectives. The normalisation applied was the division of the objective values by the mean value obtained from all points that were in the non-dominated set. This provided a single output from each MOEA that had a good balance between each objective. Due to stochasticity in the GA themselves, we ran 100 replicates of the algorithms for a fixed 50 generations and with a population size of 500. We found that this termination criterion was sufficient to find a parameter set that had converged. We quantified this by comparing the hypervolume indicator and best Euclidean distance point over different population sizes and generations [26, 27, 39] (see S1 Fig).

## Objective functions

We consider three different objectives when comparing the model output to data. Each algorithm we test uses either one (SOEA) or two (MOEA) of these. The first objective is the mean squared error (MSE) between the normalised power spectral density (PSD) in linear space:

$$MSE_{SOEA20} = \sum_{\omega_1}(\mu_{data}^{\omega_1} - \mu_{model}^{\omega_1})^2,$$

(8)

where $\mu_{data}^{\omega_1}$ gives the data PSD at frequency $\omega_1$ and $\mu_{model}^{\omega_1}$ gives the mean model PSD at frequency $\omega_1$. Here, $\omega_1$ ranges between 2 and 20Hz with a resolution of 0.125Hz [20].

For the second objective, the frequency range was widened to 2–45Hz ($\omega_2$) and the $1/f$ background noise removed by performing linear regression, using robust fitting, on the log-log spectra and subtracting this linear component [21, 40]. We then calculated the MSE difference in natural log space:

$$MSE_{SOEA45} = \sum_{\omega_2}(\log(\mu_{data}^{\omega_2}) - \log(\mu_{model}^{\omega_2}))^2.$$

(9)

The third objective, applied only in the MOEA approaches, is based on the HVG, which is a mapping from a time series to an undirected graph (network) [41]. Let $G = (V, E)$ be a graph with $V$ a set of vertices (or nodes) and $E$ a set of edges. Furthermore, let $\{x_i\}_{i = 1, \ldots, N}$ be a time series of $N$ data points. Then the HVG algorithm assigns each time point to a node in the HVG, $V = \{1, \ldots, N\}$. Nodes $i$ and $j$ are then connected by an edge if $x_i, x_j > x_k \forall i < k < j$. This means that an edge is drawn between two nodes in the graph if it is possible to trace a horizontal line between the two data points, without intersecting an intermediate point. Lacasa and Luque also proposed a useful extension of the standard HVG, known as the weighted HVG (wHVG) [42]. In this case, if there exists an edge between two nodes, then that edge is given a weighted value as the difference in amplitude between the two time points (latter time point minus former time point). See Fig 1 for an example of the calculation of the wHVG. In [28], the authors showed using information theory that the wHVG is capable of characterising dynamical systems with significantly less data than that of other commonly used metrics from HVG, such as the degree distribution. Furthermore, the authors in [43] were able to use the wHVG to classify epileptic seizures from healthy dynamics with a near 100% accuracy. In this

study, we use the two-sample Kolmogorov-Smirnov test statistic on the wHVG degree distribution to compare model and data. For $F : \mathbb{R} \to [0, 1]$ and $G : \mathbb{R} \to [0, 1]$ representing the wHVG sum of node weights cumulative distribution function of the data and model simulation respectively, and $y \in \mathbb{R}$, the third objective is defined as,

$$D = \sup_{y} \mid F(y) - G(y) \mid . \tag{10}$$

Throughout this work, to visualise differences between wHVG distributions (and also amplitude distributions), we calculate a density approximation of the distribution using 100 equally sized bins. We refer the reader to Fig 1 for the calculation of the objectives in each method.

### Jensen-Shannon Divergence

The Jensen-Shannon divergence (JSD) is a measure of the similarity between two probability distributions. It is a symmetrised and smoothed version of the Kullback-Leibler divergence. Given probability distributions $P$ and $Q$ in domain $\Omega$ and Shannon entropy $H_s(P) = -\Sigma_{x \in \Omega} P(x) log(P(x))$, then,

$$JSD(P, Q) = H_s\left(\frac{P + Q}{2}\right) - \frac{1}{2}H_s(P) - \frac{1}{2}H_s(Q). \tag{11}$$

Here, we estimate probabilities using 100 equally sized bins. We apply the JSD in two ways. First, to estimate the deviation of a parameter's marginal distribution from a uniform distribution (which approximates the prior that the optimisation started from). This enables us to quantify the practical identifiability of a given parameter, with the higher JSD implying a greater deviation from uniform and hence improved practical identifiability [20]. Second, JSD is used to estimate the discrepancy between parameter distributions recovered from control and FE subjects, with higher JSD implying a greater deviation in the recovered parameters between the cohorts.

## Results

### Comparing model dynamics recovered using different objective functions

We investigate different ways to quantify the similarity between model output and data, and explore the effect these might have on parameter mapping. Following previous studies [20], we map parameters of the Liley model [16] from resting EEG. To do this, we utilise four different global optimisation approaches, as described in Methods. Briefly, we use two different objective functions to quantify similarity in the frequency spectrum: 1) the sum of squared differences between model PSD and data PSD in the 2–20Hz range [20] and 2) the sum of squared differences between model PSD and data PSD in the log-transformed and linearly detrended 2–45Hz range (e.g. [21]). Our other objective function is the Kolmogorov-Smirnov test statistic on the wHVG degree distributions (see Methods). Our four different EAs are therefore single objective EAs using either of the frequency spectra objectives (SOEA20 and SOEA45), and multi-objective EAs combining these with the wHVG objective (MOEA20 and MOEA45). Fig 1 illustrates the calculation of these objectives.

Fig 2 shows exemplar data from two subjects, along with simulations of the model using the optimal parameters found. In the case of the MOEAs, "optimal" refers to the parameters that yield simulated objective values with the smallest Euclidean distance to the origin (see Methods). In Fig 2A it can be seen that the different algorithms yield qualitatively different dynamics, and recapitulate different aspects of the data for subject 13. For example, the SOEA20

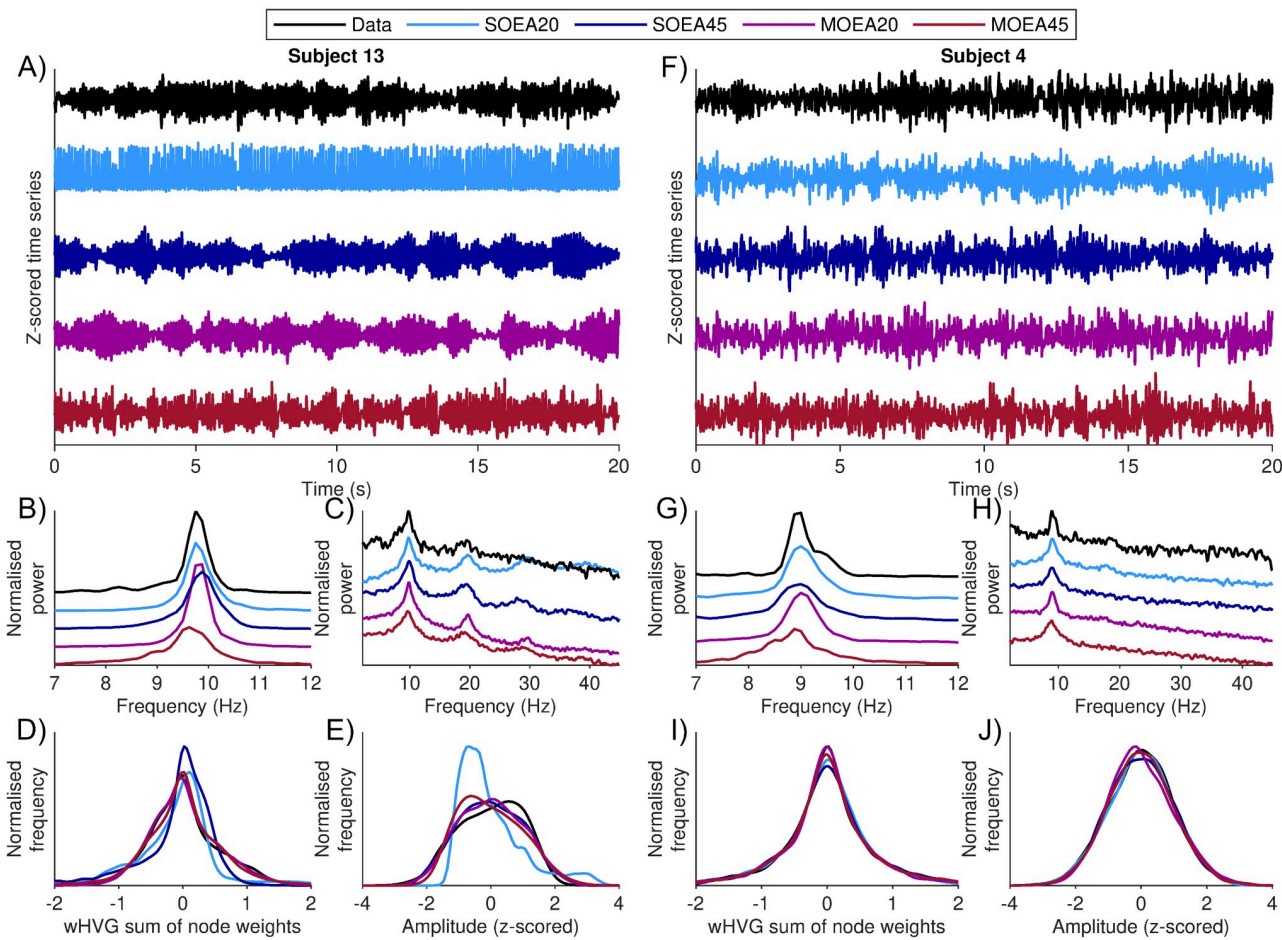

**Fig 2. Two exemplar data sets and model output using optimal parameters derived from each algorithm.** Optimal refers to the smallest Euclidean distance from the origin in objective space. A) time series from data (black) and model simulations from each algorithm (colours as per legend). The corresponding PSD for the data and each simulation is shown for B) the dominant alpha rhythm and C) the broadband spectrum. To better differentiate between the signals, the baseline of the normalised power is shown with an offset for each algorithm. D) and E) show the corresponding wHVG and amplitude distributions from each simulation against the data, respectively. In each case, these signals show a density approximation of the distribution, calculated from 100 equally sized bins (see Methods). F-J) shows the equivalent of A-E) for a second subject.

yields a very rhythmic output, and the output from the MOEA45 appears to contain more sporadic bursts than the other methods. The SOEA45 and the MOEA20 produce a waxing and waning of the alpha rhythm. Fig 2B shows a close up of the normalised PSD around the dominant alpha rhythm. The SOEA20 and the MOEA20 provide the closest match to the peak here, whereas the SOEA45 and MOEA45 have broader peaks. Examining the PSD in the broader 2–45Hz range, and using a log-transform, all algorithms except the SOEA20 have captured the over-riding "1/f" trend, and all have picked up on a second peak around 20Hz (Fig 2C). Therefore, comparing the MOEA20 to the SOEA20, the addition of the wHVG objective has led to adjustments in the power spectrum beyond the 2–20Hz range that is not explicitly quantified in the PSD objective. In the MOEA20, this is not at the expense of a reduction in the quality of fit to the predominant alpha rhythm. The SOEA45 and MOEA45, on the other hand, have produced broader and slightly shifted alpha peaks as a result of explicitly fitting to data in the higher frequencies.

Fig 2D shows that the wHVG degree distribution of the data is closely matched by the output of the MOEA20 and MOEA45, but less closely matched by the output of the SOEA20 and

SOEA45. In particular, the latter are skewed to the right and do not have as much density at high wHVG degree values. This demonstrates that the wHVG is capturing aspects of the data not accounted for by focusing on the PSD in either the 2–20Hz or the 2–45Hz range. Interestingly, Fig 2E shows that the amplitude distribution is largely conserved across all algorithms apart from the SOEA20. This confirms that the wHVG is quantifying properties of the amplitude distribution. The fact that the SOEA45 closely matches the amplitude distribution but not the wHVG distribution confirms that the wHVG is additionally quantifying aspects of the data beyond the amplitude distribution. Fig 2F–2J shows the data and model simulations for a different subject. For this subject, the alpha rhythm occurs in shorter bursts. Inspecting the time series, initially all algorithms appear to reproduce model simulations with similar dynamics to the data. However, here too the SOEA20 and MOEA20 provide better approximations to the alpha power peak in the data (Fig 2G). Furthermore, applying the MOEA20 and MOEA45 has resulted in model simulations that subtly better capture the wHVG degree distribution of the data (Fig 2I), with all algorithms largely capturing the amplitude distribution for this subject (Fig 2J). These findings are presented for all 20 subjects in S2 Fig.

To further examine the role of the wHVG, in Fig 3 we analyse shorter segments of the time series of Fig 2A. Initially, the time series of all algorithms apart from the SOEA20 appear similar (Fig 3A). However, the wHVG distribution of the SOEA45 differs from that of the data and the MOEAs. Similarly to the wHVG degree distribution of the whole segment shown in Fig 2D, the SOEA45 wHVG degree distribution is skewed to the right, and here contains most density at values between 0 and 0.5. To better understand this, in Fig 3C we re-plot the time series of the data alongside the simulations from the SOEA45 and MOEA20, with each time point colour coded according to its wHVG degree (normalised between -1 and 1, to exclude outliers). Fig 3C demonstrates that for this subject, differences in the wHVG degree distribution are largely determined by the shape of oscillations. The data and the MOEA20 contain waveforms in which the rise phase has very high wHVG degree (white areas in time series of Fig 3C). In contrast, the SOEA45 has waveforms in which the decay phase of the oscillation has very low wHVG degree (dark areas in the time series of Fig 3C). The data and MOEA20 also contain some periods of more symmetric oscillations, for which the SOEA45 does not. In summary, for this subject, applying the MOEA20 algorithm has maintained a good approximation to the alpha rhythm peak (in contrast to the MOEA45) whilst avoiding dynamics that are too periodic (in contrast to the SOEA20) or of a different shape to the data (in contrast to the SOEA45).

Fig 4 summarises these findings across all 20 subjects. It shows the distribution of differences in the normalised median objective values, considering the MOEA20 as a reference point. Fig 4A shows that the MOEA20 and MOEA45 clearly perform best in approximating the wHVG of the data across all subjects. The MOEA20 also better approximates the PSD in the 2–20Hz range compared to the SOEA45 and MOEA45; the fit of the MOEA20 to the alpha peak in the 2–20Hz range is indistinguishable from the SOEA20 in the majority of subjects (Fig 4B). Taken together, this indicates that the MOEA20 is performing well at capturing an optimal trade-off between PSD fit to the dominant rhythm in the alpha range and the wHVG. In performing this trade-off, the MOEAs also produce parameter values for which simulations approximate the 2–45Hz log-transformed frequency spectrum well. This can be seen in Fig 4C, in which the SOEA45 performs only marginally better than the MOEAs on the whole. We note that the use of log-transformed and detrended 2–45Hz band as an objective in the SOEA45 yields outliers with very poor fits to the dominant alpha peak, as seen in the example of S3 Fig. S4 Fig shows an alternative representation of these findings, demonstrating that the MOEA20 produces simulations with the lowest Euclidean distance from the origin in either 2 or 3-dimensional objective space. In addition, S5 Fig shows that, in general, the results of the

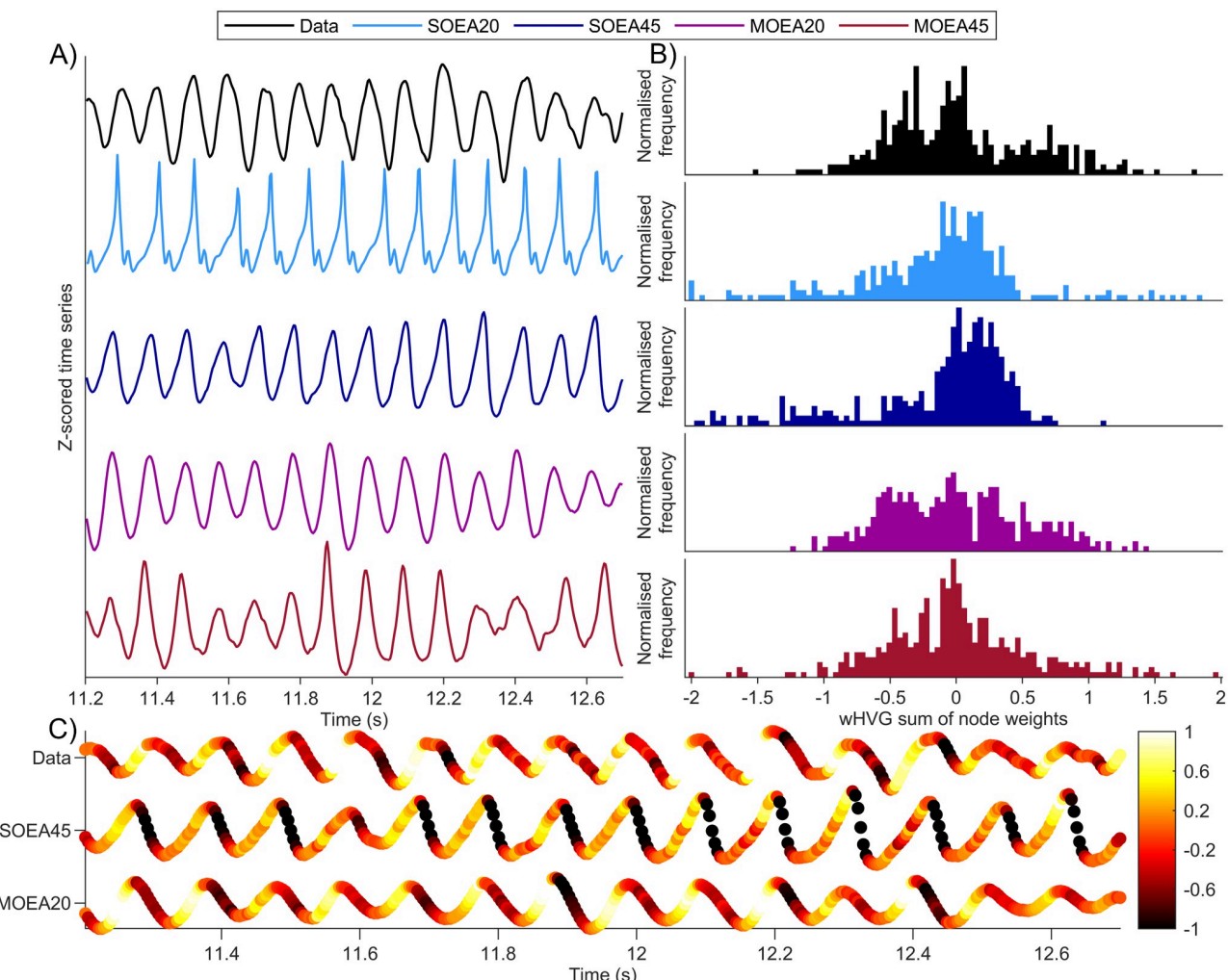

**Fig 3. Illustration of wHVG distributions derived from the data presented in Fig 2A.** A) 1.5s segments of data and model simulations at the optimal parameter set (colours as per legend). Optimal refers to the smallest Euclidean distance from the origin in objective space. B) the corresponding wHVG distributions derived from the time series segments in A). These histograms give the wHVG frequency from 100 equally sized bins (see Methods). C) the time series from the data, SOEA45 and MOEA20 simulations, as shown in A), with time points (i.e. wHVG nodes) coloured according to their node weight (normalised between -1 and 1 to remove outliers).

MOEAs better approximate the amplitude distribution and the Hurst exponent than the SOEA20 and SOEA45.

## Comparing parameter values recovered using different objective functions

We have shown that the choice of objective function can alter the dynamics that are generated by the model. These changes in dynamics can be major (such as the SOEA20 producing spikes, as shown in Fig 2) or more subtle (such as the difference in waveform between the MOEA20 and SOEA45 encapsulated by the wHVG, as shown in Fig 3). A natural question that arises is, to what extent this will affect the values of the parameters that are recovered. Fig 5 demonstrates that the differences can be substantial. These results are derived from each of the subjects analysed in Fig 2. For the first subject, Fig 5A shows that the MOEA20 and MOEA45 algorithms have resulted in a narrowing of the distributions of some of the parameters compared to the SOEA approaches. Specifically, $\gamma_e$ and $\gamma_i$ are more constrained to the lower end of

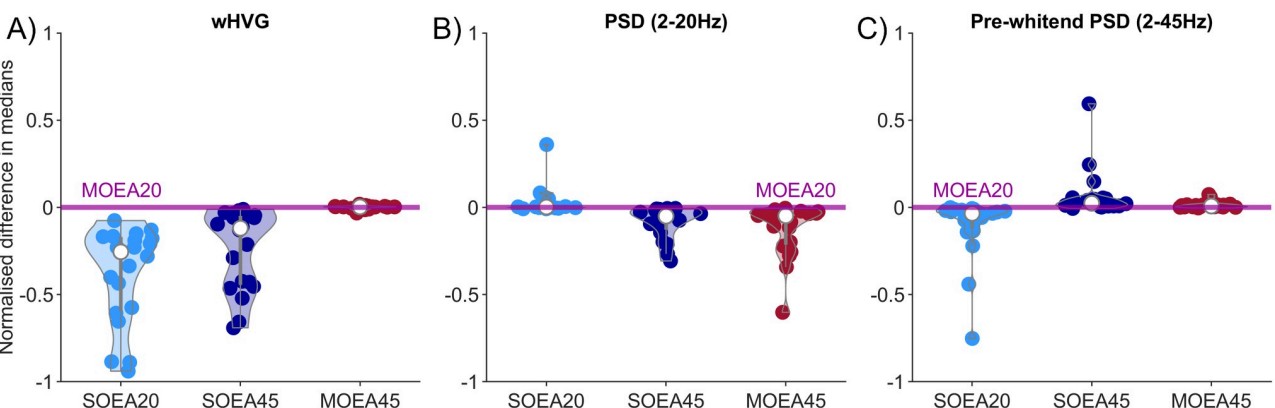

**Fig 4. Normalised median objective scores from all fits to data and across each algorithm, with the MOEA20 as a reference.** Each scatter point represents a median objective score from a single subject. The median is calculated from the distribution of normalised Euclidean distances from the origin. This is shown for the A) wHVG B) PSD 2–20Hz and C) log-transformed PSD 2–45Hz objectives. A distribution above (below) the MOEA20 line indicates a better (worse) score was obtained from that algorithm on that objective, compared to the MOEA20.

the ranges explored, whereas $\sigma_e$ is more constrained to the higher end of the range explored. Conversely, for the second subject analysed, Fig 5B shows that using the different algorithms can result in vastly different recovered parameter values. This is particularly apparent for parameter $\gamma_e$. Here, the SOEA20 yields $\gamma_e$ values across the whole range, SOEA45 towards the upper end, MOEA20 predominantly at the lower end and MOEA45 mainly around the middle of the range. For this subject, the SOEA20 approach additionally deviates from the other algorithms for parameters $\gamma_i$ and $\sigma_e$.

To visualise and quantify differences with respect to the full parameter space, we calculated the normalised Euclidean distance between parameter sets recovered from each algorithm in the full space and then map this to 2 dimensions using multi-dimensional scaling. Parameters in the same location in the full space would map to the same position in this 2-d space. Four pairwise algorithm comparisons for the position of the $N = 100$ repeats in this space are colour coded and shown in Fig 5C and 5D. The algorithms partially separate into clusters, although there is a large degree of overlap. We quantified this effect by calculating the silhouette score, considering points from different algorithms as different clusters. The scores are shown in the titles of the subplots for the 2-d space comparison, with the values in the full space given in brackets. For the first subject analysed, the largest silhouette score is for the MOEA20 compared to the SOEA20 (Fig 5C). For the second subject analysed, Fig 5D shows all the algorithms largely separate into distinct clusters with all silhouette scores in the 2-d space $\geq 0.24$. We followed this same procedure to quantify differences in the location of recovered parameter sets produced by each of the 4 algorithms across all 20 control subjects. The silhouette score from the 2-d space, for each comparison, is shown in Fig 6, along with the means over all 20 subjects. In addition, S6–S9 Figs give the parameters in the 2-d space across all the subjects that these scores were derived from. The average silhouette scores are around 0.18 for all comparisons considered, except for the MOEA20 compared to the MOEA45, which is approximately 0.09. Considering the silhouette scores in the subjects analysed in Fig 5, we see that even for low values of the silhouette score, large differences in the distributions of one or more parameters can exist. For example, as shown in Fig 5C, the MOEA20 compared to the SOEA45 has a below mean silhouette score of 0.15. However, it can be seen in Fig 5A that for the MOEA20, in comparison to the SOEA45, the parameter $\sigma_e$ alone is much more constrained to the higher end of the range explored.

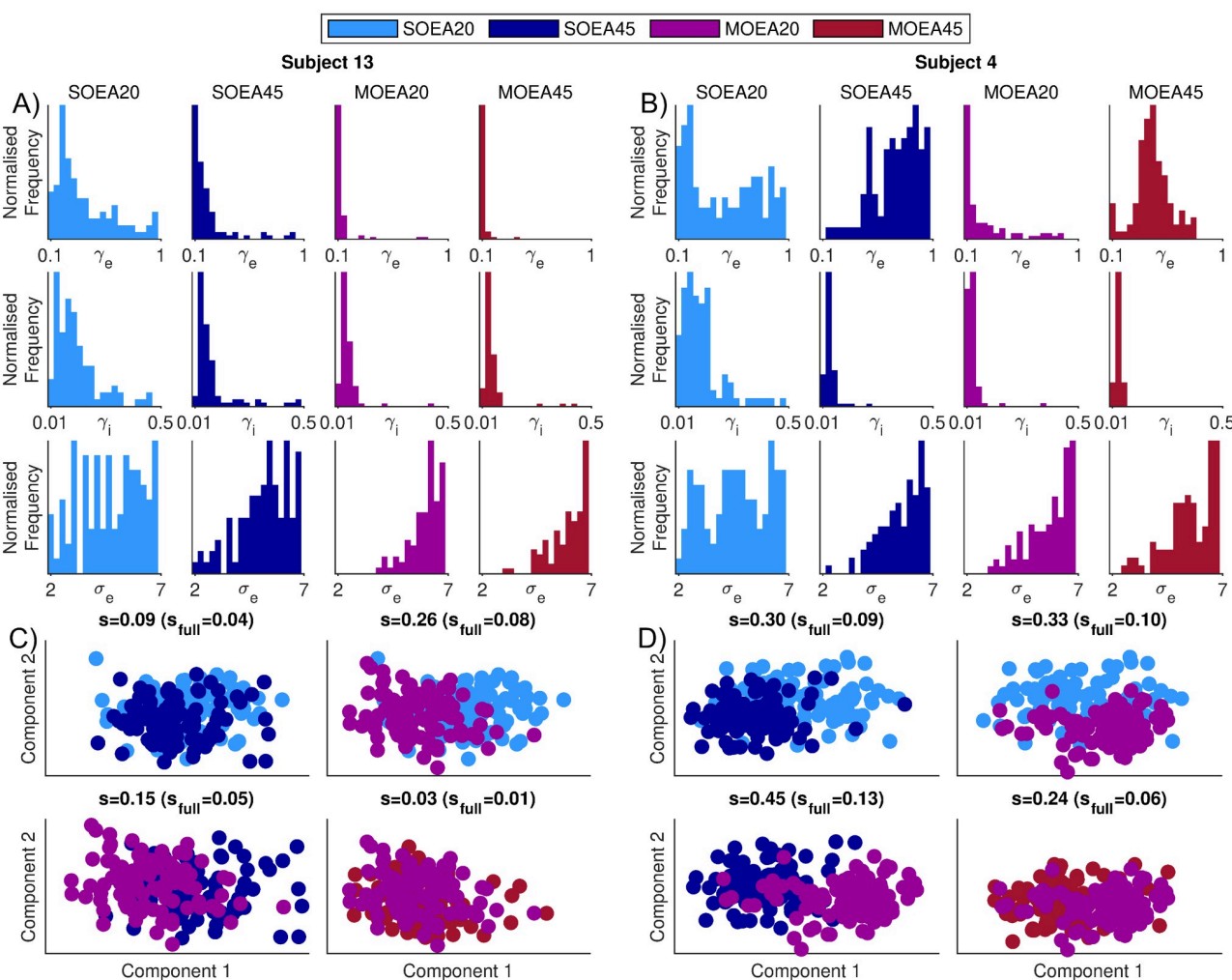

**Fig 5. Example distributions of parameter values recovered from two subjects, using each algorithm.** A) and B) show the distribution of 3 parameters ($\gamma_e$, $\gamma_i$ and $\sigma_e$) that were recovered from subjects 13 and 4, respectively. These were the same two subjects analysed in Fig 2. The parameter bounds are set to those used in the optimisation (see Table 1). For these subjects, C) and D) show the projection of each parameter set into the 2-d space found by applying a multi-dimensional scaling to the full parameter space. Each scatter point represents the optimal position in parameter space found by a single run of the optimisation algorithm. Optimal refers to the smallest Euclidean distance from the origin in objective space. The separation of these points in 2-d space is quantified by the silhouette score, with the silhouette score in the full space given in brackets.

We next examined the distributions of parameters recovered from each algorithm, across all subjects combined. Histograms for all parameters are shown in S10–S13 Figs. On the whole, we found that for many of the parameters, feasible values were present across the whole range explored. However, many parameters also displayed modes, i.e. higher densities at certain values. To quantify this effect, we calculated the JSD between the distribution of parameters recovered from control subjects, compared to a uniform distribution (see Methods). We found that in at least one of the algorithms, 5 of the 22 parameters had distributions significantly different from uniform. These parameter distributions are displayed in Fig 7. It can be seen that the synaptic time scale parameters, $\gamma_i$ and $\gamma_e$, have the largest deviation from uniform, and are peaked at low values. However, for the SOEA20 and SOEA45, the distribution of these parameters contain additional density at high values. This is reflected in the lower values of the JSD for these algorithms. The distributions of the other parameters in Fig 7 in general can also

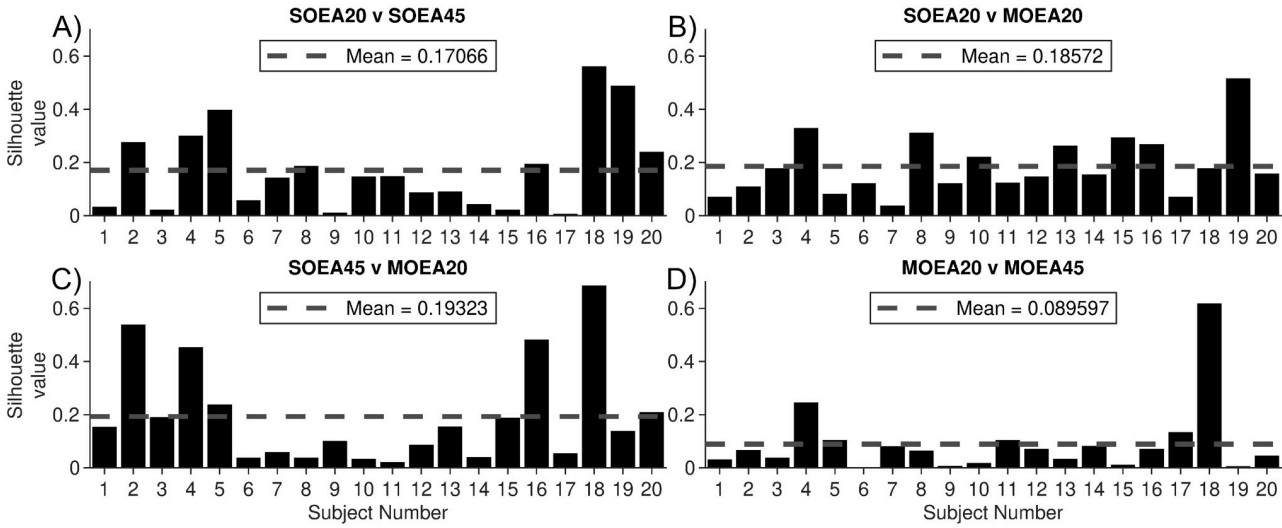

**Fig 6. Silhouette scores quantifying differences in parameters recovered from different algorithms.** The parameter values are mapped via multi-dimensional scaling to a 2-d space and silhouette scores calculated for pairs of algorithms. A) SOEA20 v SOEA45, B) SOEA20 v MOEA20, C) SOEA45 v MOEA20 and D) MOEA20 v MOEA45 comparisons. In each case, horizontal dashed lines give the mean silhouette score across subjects for the corresponding pairwise comparison.

be seen to have been somewhat "filtered" or "trimmed" by each of the MOEA approaches, and the MOEA20 approach in particular. This is quantified by the MOEA20 having a significantly larger JSD than both the SOEA20 and SOEA45 for parameters $\gamma_e$, $\gamma_i$, $\sigma_e$ and $p_{ee}$. This indicates that, in addition to the MOEA20 recovering parameter values that are different from the other algorithms, it has the added benefit that these values are less dispersed in parameter space.

Having quantified differences in the dynamics recovered under different objective functions, we explored whether these were reflected in the nature of the invariant objects that organise these dynamics in phase space. For simplicity, and since most relevant dynamics resulted from the deterministic part of the model being at steady state, we focused on recovered parameter sets for which the model had a stable fixed point. Fig 8 shows the distribution of real parts of the dominant eigenvalue for these fixed points, across each algorithm. It can be seen that the distributions for the MOEA20 and MOEA45 are much more constrained than those of the SOEAs, and reside at values closer to zero. Thus, an additional effect of the MOEA approaches is to recover fixed points that attract nearby trajectories more slowly and are potentially more proximal to bifurcations. We note that for 2 of the 20 subjects studied, the majority of parameter sets recovered placed the deterministic part of the model in a limit cycle regime for all algorithms. Limit cycle dynamics are only possible when simulating the full nonlinear model. The type of attracting state reached for a given parameter set was quantified by the eigenvalue spectrum. The proportion of parameter sets giving rise to noise-driven fixed points and noise-driven limit cycles are shown in Fig 9A. Fig 9B and 9C provide an example model simulation in phase space from an attracting noise-driven fixed point and an attracting noise-driven limit cycle, respectively.

Additionally, we tested the different algorithms on signals simulated by the model. This allowed the distribution of parameters recovered from the algorithms to be compared to a ground truth value. S14 Fig shows, across the same parameters as those considered in Fig 7, the Euclidean distance to the ground truth from 3 simulated time series. S14 Fig also shows the histograms of parameter values from one of these simulated time series. Across all 3 simulated time series, compared to the other algorithms, we found that the MOEA20 approach was either

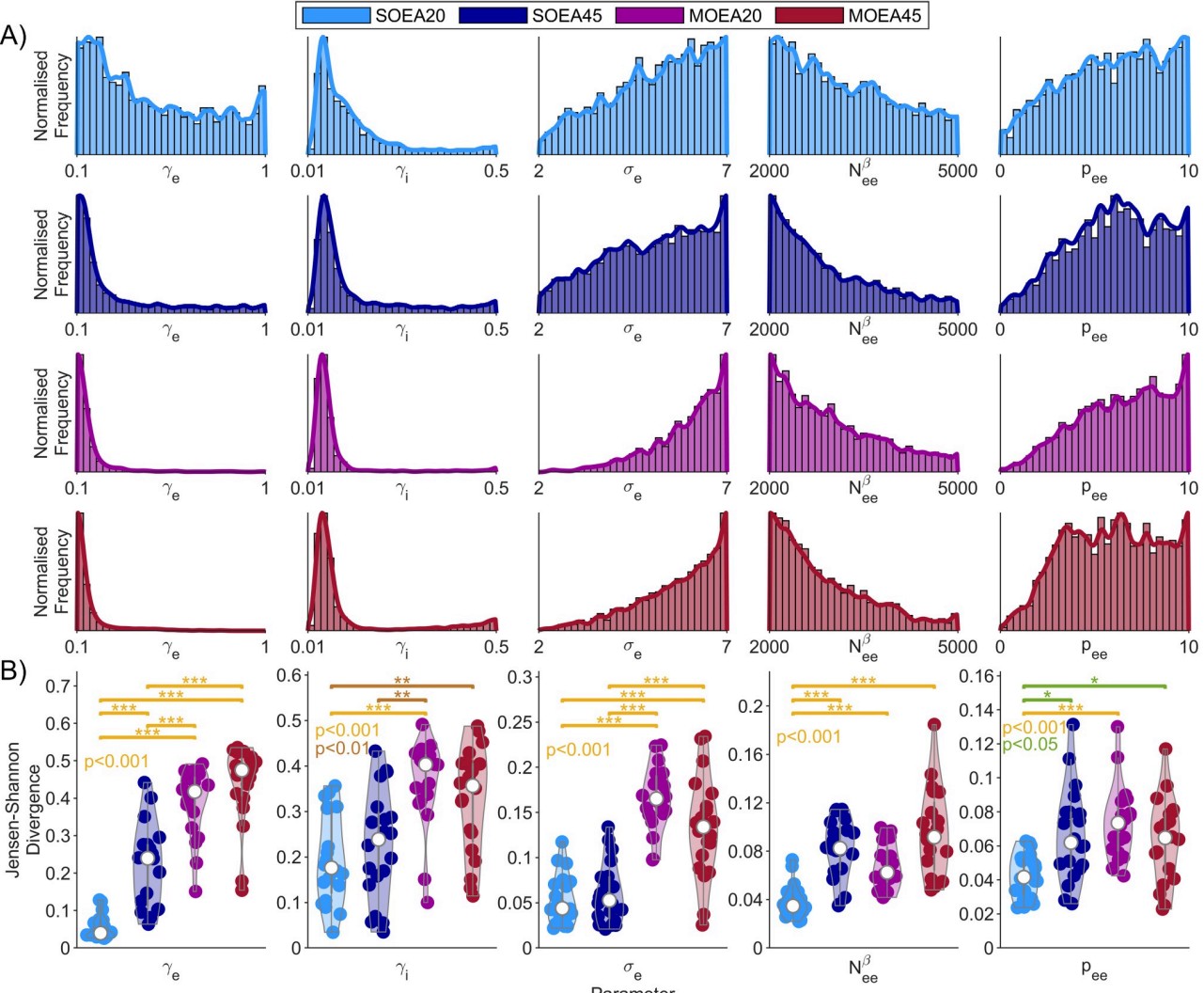

**Fig 7. Parameter distributions recovered from all control data sets.** A) shows the distributions of 5 of the parameters that were recovered from 100 repeats of each algorithm on all subjects (coloured as per the legend). These 5 parameters represent the biggest deviation from a uniform distribution in at least one of the algorithms (see Methods). The parameter bounds in these plots are set to those used in the optimisation (see Table 1). B) shows the JSD of the marginal distributions, when compared to a uniform distribution. These are shown as violin plots over subjects. P-values were obtained from a Mann-Whitney U test, with Bonferroni correction.

insignificant from, or significantly closer to, the original ground truth value used to run the simulated data.

## Using MOEAs to explain alpha power shifts in epilepsy

Our results thus far show that the MOEA20 is better than the other algorithms at capturing relevant features of the data. We exclude the SOEA20 due to the spiking limit cycle behaviour it can give rise to (Fig 2); the SOEA45 due to it giving rise to waveforms of a different shape to the data (Fig 3) and the MOEA45 as it detracts from dominant alpha rhythms (Fig 2). Essentially, the MOEA20 provides the best algorithm to capture the alpha rhythm PSD plus the waxing and waning, as well as the shape of the waveform (as shown in S4 Fig). We therefore focus on the MOEA20 and apply it to the problem of understanding differences that have been

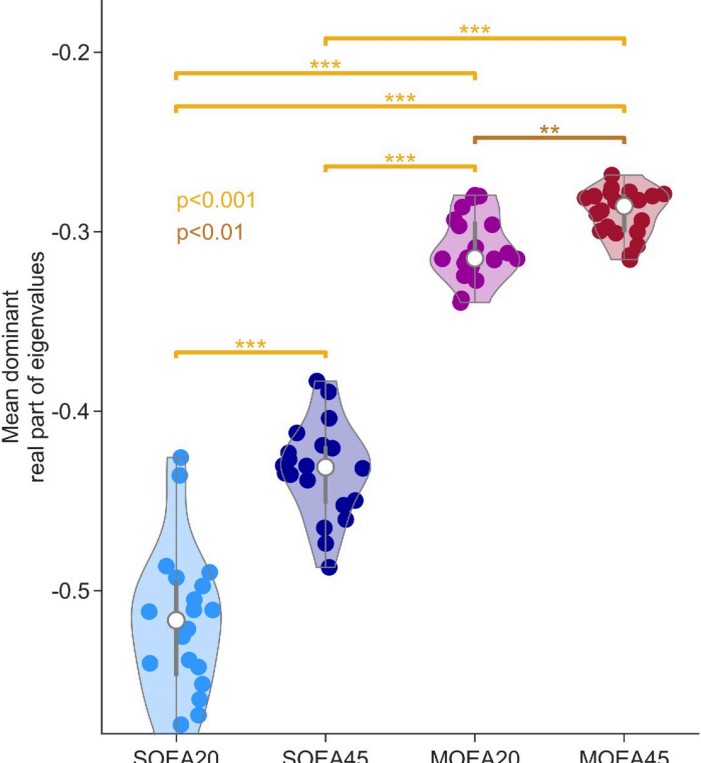

**Fig 8. Distributions of the mean real part of the dominant eigenvalue for each subject across algorithms.** Each point is derived from the optimal parameter set recovered from a single data set. Optimal refers to the smallest Euclidean distance from the origin in objective space. Parameters yielding unstable fixed points are excluded. P-values were obtained from a Mann-Whitney U test, with Bonferroni correction.

observed in the dynamics of resting EEG in people with epilepsy versus healthy controls [35]. In particular, a shift in the alpha power of epileptic patients to lower frequencies has previously been reported [35]. Fig 10A and 10B show the PSD in the 2–20Hz range and the wHVG distribution for controls and people with FE, respectively. The FE shift towards lower frequencies can be seen, as can a small increase in the peak of the modal wHVG for the FE subjects. Fig 10C and 10D show the PSD and wHVG distributions of simulations of the model at parameter sets recovered via the MOEA20, respectively. It can be seen that the simulations accurately recreate the power spectra and wHVG of both cohorts and the differences between them.

Using the MOEA20, we next compared the distribution of parameter values that were recovered from the control and FE data. The comparison of all parameters can be seen in S15 Fig. It is apparent that many of the parameters have values across the full range explored, and in many cases, there are no differences between the distributions derived from people with epilepsy and healthy controls. Of notable exception is the parameter $\Gamma_e$, which quantifies the mean excitatory synaptic gain. We found the density of this parameter to be shifted to higher values in healthy subjects compared to people with epilepsy. In order to quantify any other systematic differences in the distributions of parameter values between the cohorts, we calculated the effect sizes (Cohen's d) and JSD between the marginal distributions. Fig 11 shows the distributions of the 4 parameters with the largest differences according to both measures. In addition to $\Gamma_e$, these were the inhibitory postsynaptic rate constant, $\gamma_i$, the reversal potential for the membrane on inhibitory neurons, $h_i^{eq}$ and the excitatory postsynaptic rate constant, $\gamma_e$.

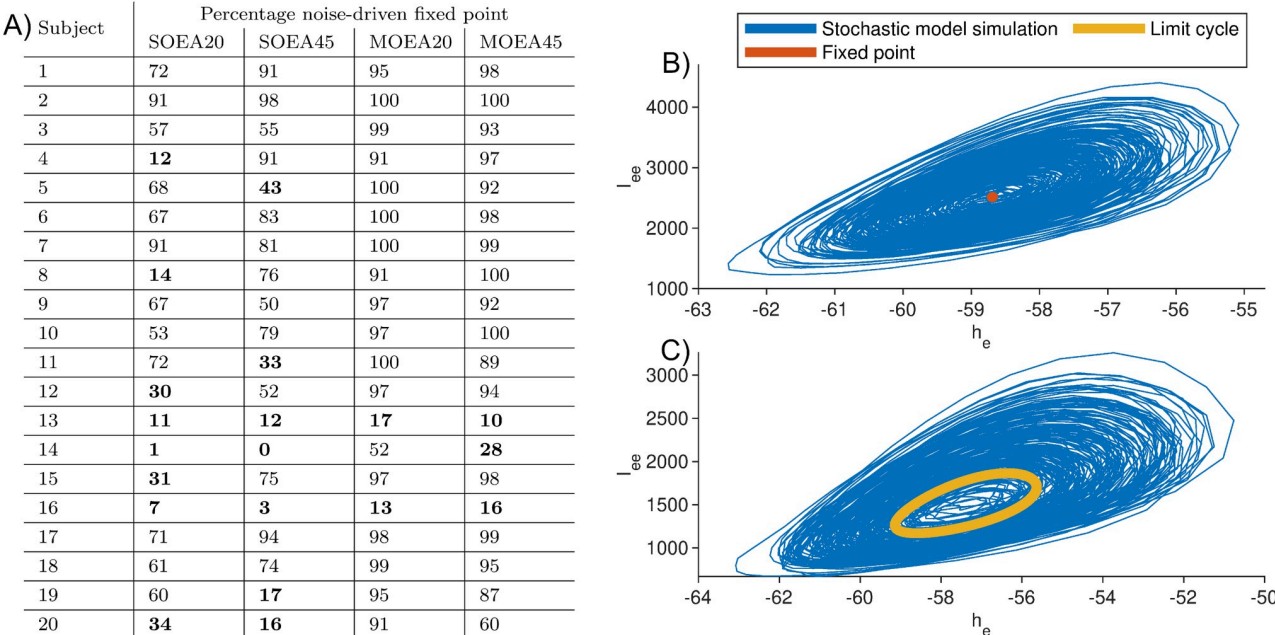

| A) Subject | Percentage noise-driven fixed point | | | |
|---|---|---|---|---|
| | SOEA20 | SOEA45 | MOEA20 | MOEA45 |
| 1 | 72 | 91 | 95 | 98 |
| 2 | 91 | 98 | 100 | 100 |
| 3 | 57 | 55 | 99 | 93 |
| 4 | **12** | 91 | 91 | 97 |
| 5 | 68 | **43** | 100 | 92 |
| 6 | 67 | 83 | 100 | 98 |
| 7 | 91 | 81 | 100 | 99 |
| 8 | **14** | 76 | 91 | 100 |
| 9 | 67 | 50 | 97 | 92 |
| 10 | 53 | 79 | 97 | 100 |
| 11 | 72 | **33** | 100 | 89 |
| 12 | **30** | 52 | 97 | 94 |
| 13 | **11** | **12** | **17** | **10** |
| 14 | **1** | **0** | 52 | **28** |
| 15 | **31** | 75 | 97 | 98 |
| 16 | **7** | **3** | **13** | **16** |
| 17 | 71 | 94 | 98 | 99 |
| 18 | 61 | 74 | 99 | 95 |
| 19 | 60 | **17** | 95 | 87 |
| 20 | **34** | **16** | 91 | 60 |

**Fig 9. Comparison of the attracting state reached by model simulations.** A) gives the percentage of model simulations that consisted of noise-driven fixed dynamics (as opposed to noise-driven limit cycle dynamics) for each algorithm and across all subjects. Subject-algorithm combinations that were predominantly described by noise-driven limit cycle dynamics are given in bold. B) gives an example simulation in phase space of noise-driven fixed point dynamics. Here, due to the stochasticity, the model oscillates around a stable fixed point. C) gives an example simulation in phase space of noise-driven limit cycle dynamics. In this case, the stochastic model simulations oscillate around a stable limit cycle.

To understand the effect that these parameter changes would have on the underlying activity of the modelled neural masses, we calculated the membrane potential and firing rate that were obtained in simulations at the parameter values recovered from controls and FE subjects. S16 Fig shows that the differences in recovered parameters map onto a significant increase in both the membrane potential and firing rate in controls, in comparison to FE, for both the excitatory and inhibitory neural populations.

## Optimising model dynamics to spike-wave discharges

Having successfully applied the MOEA20 algorithm to resting EEG, we sought to test whether it would be a useful tool to uncover the presence of nonlinear dynamics in the model. We therefore applied the MOEA20, and additionally the SOEA20, SOEA45 and MOEA45 for comparison, to a time series containing spike-wave discharges (SWDs), which is an archetypal, nonlinear epileptiform rhythm. Fig 12 shows that the MOEA20 is able to find parameter values for which the model generates an approximation to the SWD, in terms of visual appearance, PSD and wHVG. This is in contrast to the single objective algorithms we tested, which can match the PSD but do not yield an appropriate waveform, as can be seen in Fig 11. We note that this analysis was performed using the deterministic (noise-free) simulation of the model. S17 Fig shows the distribution of parameters recovered from the SWD using the MOEA20 approach.

## Discussion

In this study, we presented a new approach for recovering parameters of neural mass models (NMMs) from data. We combined the simulation of nonlinear NMMs with global exploration

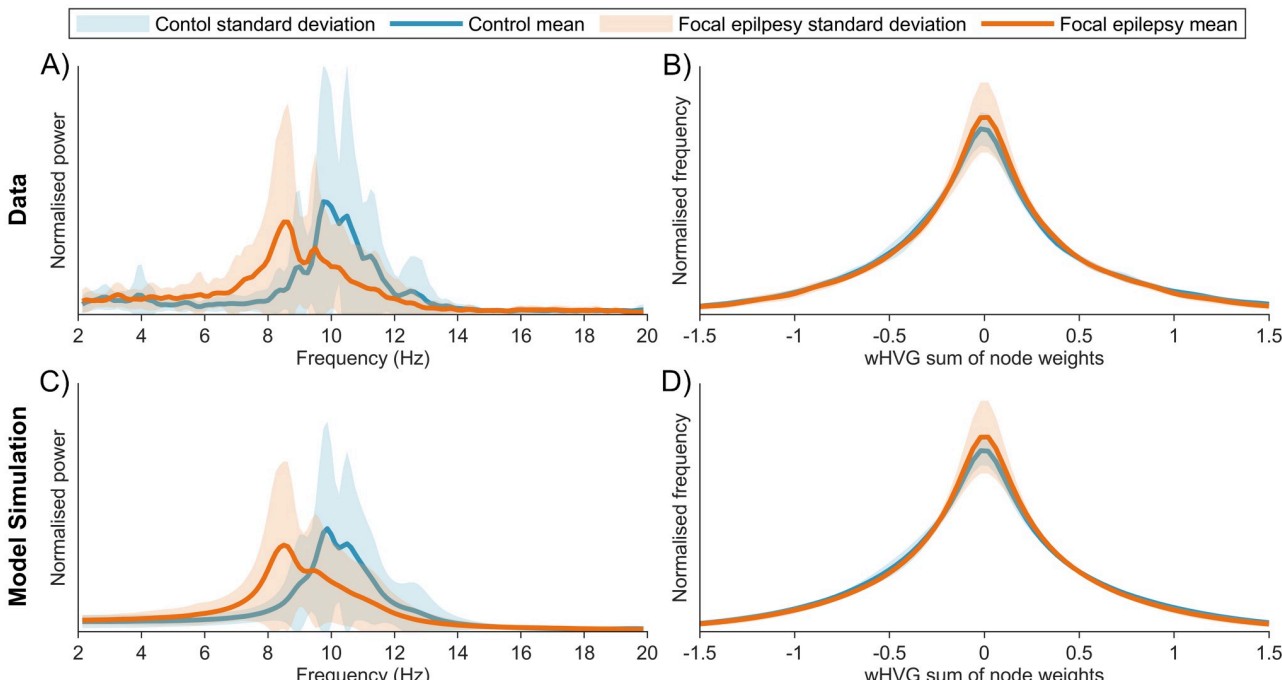

**Fig 10. Comparison of data and optimal model simulations for control and FE subjects.** A) shows the mean and standard deviation of the PSD across control and FE subjects. B) shows the mean and standard deviation of the wHVG distribution across control and FE subjects. C) and D) give the PSD and wHVG distribution from model simulations at optimal parameter values. Optimal refers to the smallest Euclidean distance from the origin in objective space. Colours as per legend.

of parameter space via a MOEA and used the wHVG as an additional measure to quantify the similarity of model output and data. Comparing this approach to SOEAs, we highlighted that different choices of objective function can affect the model dynamics and parameter values that are recovered. For example, we demonstrated that substantial differences in parameter values could emerge from the same data, depending on whether the PSD was quantified in the 2–20Hz or the log-transformed 2–45Hz band, which are different strategies that have previously been employed [20, 40]. We found that the addition of the wHVG and the use of the MOEA was able to restrict the kind of dynamics that were recovered; limit cycles were ruled out where not appropriate, and fixed points with a narrow range of dominant eigenvalue real parts were preferentially found. This was accompanied by parameters being mapped to different regions of parameter space. These are important findings because quantifying differences between model output and data is integral to any parameter inference method, including probabilistic frameworks like dynamic causal modelling (DCM) or Kalman filtering. In such methods, recursive updates to posterior distributions depend upon the difference between model predictions and data [44, 45], which could evolve differently if the model and data are transformed, for example, between the time and frequency domain [46]. Care must therefore be taken to caveat inferences, or to check whether they are robust to methodological choices.

Qualitatively different dynamics can yield the same PSD, for example, due to differences in amplitude or phase distributions [23]. Thus, when comparing nonlinear models to data via the PSD, spurious limit cycle dynamics may be recovered, as shown in Fig 2. We found that this issue was mitigated by the addition of the wHVG objective, since it constrained the amplitude distribution and shape of the dynamics of the model. Additionally, we found that the wHVG objective can help to capture subtleties in the waveform of the data, which were not captured

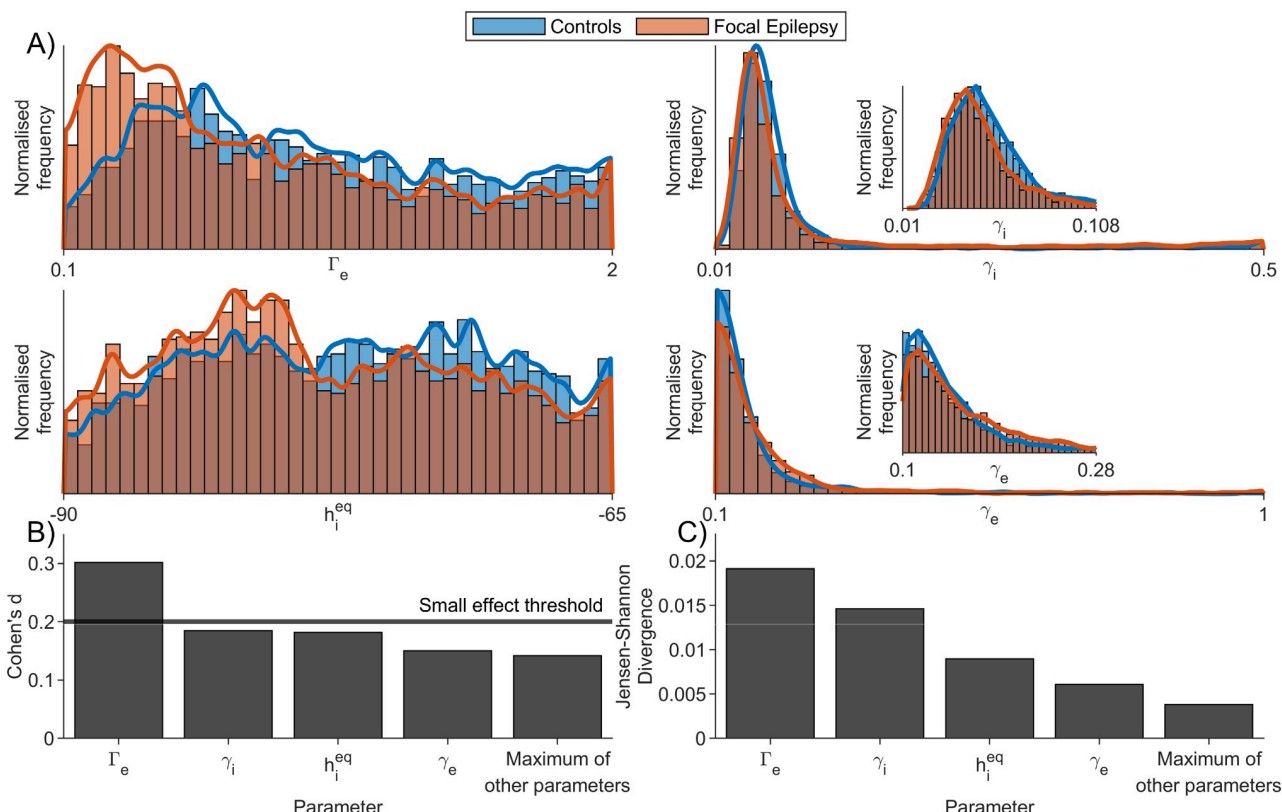

**Fig 11. Comparison of parameter values recovered from control and FE data using the MOEA20 algorithm.** A) shows the distribution of 4 parameters that were recovered from control and FE data (colours as per legend). The bounds of these parameters were set to those used in the optimisation (see Table 1). The 4 parameters chosen are those with the largest Cohen's d effect size for differences between the groups, as shown in B). C) shows the JSD between the parameter distributions recovered from each cohort. This represents an alternative measure of dissimilarity between the distributions. The comparison between all parameters in the model can be seen in S15 Fig. The insets for parameters $\gamma_i$ and $\gamma_e$ provide a close up of the most populated part of the distributions, showing only the bottom 20% of each parameter range.

when using the PSD alone (Fig 3). In 2 out of 20 subjects, we found nonlinear dynamics in the form of noise-perturbed limit cycles (rather than noise-driven fixed points) to be the most appropriate model. This is in line with previous findings that some recordings of alpha rhythms show features of nonlinear dynamics [23]. [23] estimated the percentage of alpha rhythm recordings showing nonlinearity to be 1.25%, which is lower than the 10% found here. However, they studied more data (4 epochs from each of 60 subjects), and shorter time segments (2.5s). To quantify nonlinearity, [23] used phase-randomised surrogates derived directly from time series and used a NMM to demonstrate that limit cycle dynamics were correctly assigned as nonlinear by their method. The algorithm we presented herein provides an alternative, mechanistic approach for detecting nonlinearities. The model with nonlinearities is fit directly to the data, with the possibility that stable steady states or limit cycles provide the best representation of the data. It can also be used to generate artificial EEG data with specific properties of interest.

Another potential use for our method is to search for the presence of different kinds of dynamics in NMMs. We demonstrated this by applying our method to SWD data. Notably, we were able to find relevant SWD dynamics in the Liley NMM, which to our knowledge have not been previously reported. In modelling studies of brain rhythms, traditionally, the dynamics of NMMs have been explored by varying a few parameters around default values in simulation

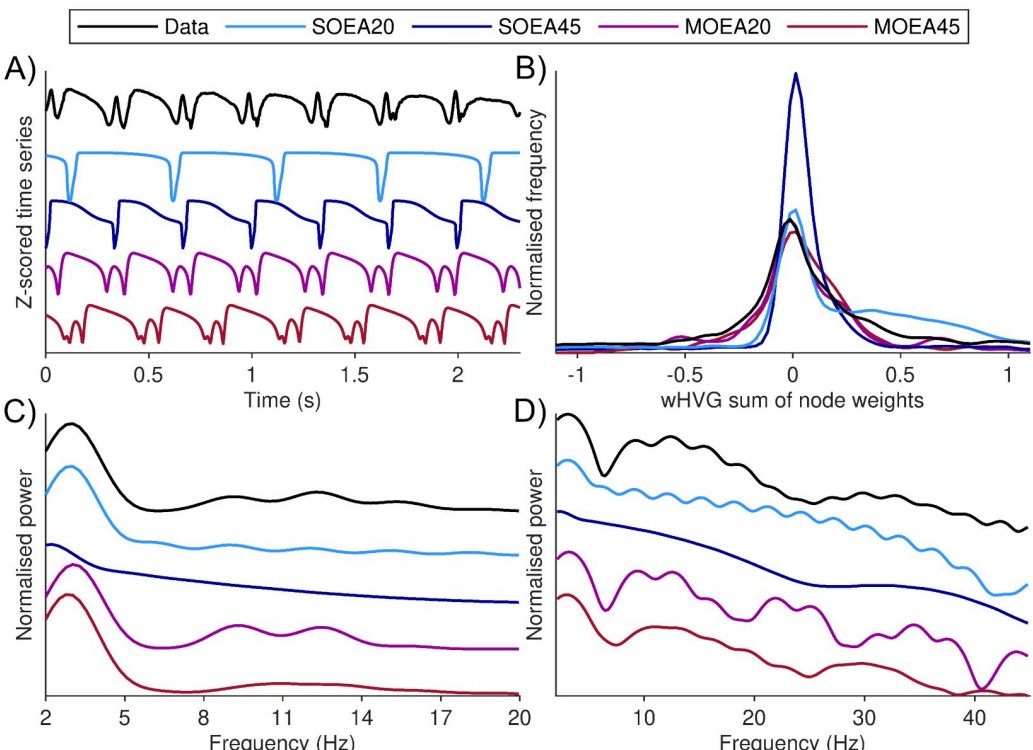

**Fig 12. MOEA algorithms can uncover the presence of a SWD rhythm in the model.** A) shows a segment of EEG displaying spike-wave dynamics, along with simulations of the model at the optimal parameters recovered using each algorithm (colours as per legend). Optimal refers to the smallest Euclidean distance from the origin in objective space. B) shows the corresponding wHVG distribution for the data and model simulations using the optimal parameters recovered from each algorithm. These signals show a density approximation of the distribution, calculated from 100 equally sized bins (see Methods). C) and D) show the corresponding PSD in the 2–20Hz and log-transformed 2–45Hz range, respectively. To better differentiate between the signals, the baseline of the normalised power is shown with an offset for each algorithm. The SWD data was recorded from frontal scalp EEG electrodes from a patient with absence epilepsy at Inselspital Bern, Switzerland.

studies or using numerical continuation and examining the emergence of instabilities or different kinds of limit cycles [4, 9, 12, 13, 17, 18, 47], though more global approaches have also been used [8, 15]. Our approach is less constrained since we allow all parameters to vary, and the model output is compared quantitatively to the data as part of the algorithm. It could therefore be a useful tool for the development of NMMs in the future, since the repertoire of dynamics of any (new) model can be readily checked against the repertoire of dynamics of the brain in its natural states or in experimental manipulations (which are important for the development of new models [10]). Using a global approach allows us to comprehensively map the presence of different kinds of rhythms, and examine their relative position in parameter space. In future, we believe this approach could help to uncover a repertoire of possible mechanisms by which NMMs can transition from a resting to pathological state, thus providing insight into the mechanisms of epilepsy.

We consider four different objective functions when comparing model output to data. The choice of objective function to use will ultimately be informed by the properties of the EEG one is interested in recapitulating in the model. For the data we analysed, the MOEA20 provided the best way to recreate the PSD of the alpha rhythm, whilst also simulating realistic characteristics in the temporal domain, such as fluctuating amplitudes and the shape of oscillations. We therefore focused on applying the MOEA20 method to study the mechanisms

underlying differences seen in the frequency spectrum of resting EEG in people with epilepsy versus healthy controls [35]. Analysing the differences in terms of univariate parameter distributions, we found that one parameter in particular stood out as having a visibly different distribution between the two cohorts: $\Gamma_e$, which quantifies the average synaptic gain for the excitatory population. Interestingly, the bulk of this distribution was found to be shifted to lower values in people with epilepsy, indicating a reduced excitatory synaptic effect in that cohort overall, *at the macroscopic level of the neural mass*. It is interesting and non-trivial that a synaptic gain parameter would be implicated in the shifts in the power spectrum observed. Since the peak alpha rhythm exists at lower frequencies in people with epilepsy, it might be expected to be explained in the NMM by an adjustment to the time scale parameters. Indeed, we did find a difference in the parameters $\gamma_i$ and $\gamma_e$. However, the effect size of these was smaller than that of $\Gamma_e$. The differences that we observed in parameter values between the two cohorts resulted in a net increase in the membrane potential and firing rate in controls, compared to people with epilepsy. It is interesting to be able to map dynamics recorded in the resting state, without the presence of epileptiform activity, onto mechanisms that capture notions of "excitability". Due to small differences found in the average synaptic gain for the inhibitory population, these differences observed in $\Gamma_e$ map onto a lower resting state excitation/inhibition ratio in FE subjects, compared to control. Interestingly, other work has provided evidence for an increase in this ratio in epilepsy patients during the start of a seizure (for example, see [48]). Mechanistically, we therefore speculate the presence of a homeostatic mechanism in the brains of people with epilepsy that leads to reduced excitability in the non-seizure state. This could be innate or due to antiseizure medication and would be an interesting avenue for future investigation.

A potential issue when mapping parameters of models of biological systems is that they are inevitably non-identifiable, or sloppy [49]. This means that many different parameter values, in potentially disparate regions of parameter space, could equally well account for the dynamics observed in data. Here, we demonstrated that the practical identifiability of model parameters is dependent upon the method used to quantify the similarity between data and model output (Fig 7). In particular, we found $\gamma_e$ as well as $\gamma_i$ to be constrained to a small range of values when using the MOEA approaches, as opposed to the other methods. Our SOEA20 uses the same objective function employed by [20], in which $\gamma_i$ was found to be the only identifiable parameter. The distributions of some other parameters also deviated further from uniform when using the MOEAs, compared to the SOEA20. Our findings, therefore, suggest that NMMs may be more identifiable than previously reported [20, 50].

In addition, we applied our algorithms to model simulations, in order to understand the relationship of recovered parameters with respect to a ground truth. As described above, the notion of "ground truth" should incorporate large regions of parameter space. That is, as we demonstrated in S14 Fig, given a choice of model parameters to simulate from, the optimisation algorithms will recover large regions of parameter space that yield equivalent, or quantitatively similar, dynamics. Thus the "ground truth" may not be a single value, rather a non-trivial set, or collection of values in a high dimensional space. It is important to be able to map these regions in order to understand the repertoire of dynamics of NMMs. The algorithms we provided herein offer the means to do this, and in particular the MOEA20 provides a more accurate map.

Any inference made from models of course depends on the choice of model, and several different types of NMMs exist (for example, see [1]). Although we focused on the Liley model here, future work will examine the application of the MOEA approach to other NMMs, such as those that contain different types of inhibitory interneurons [4, 12, 51]. We believe the

MOEA proposed could provide a better data-driven quantitative analysis of the importance of including different mechanisms in NMMs.

The problem of deciding how to compare model output to data is non-trivial. Future work will further probe the potential benefits of comparing model output to data in different ways. One benefit of using the wHVG is that it opens up the possibility of quantifying time series using a suite of different methods from graph theory [30, 52]. It will be interesting to examine how measures relating to clustering or centrality, for example, affect the results of mapping model parameters from data. The wHVG algorithm has also previously been successfully used to delineate between different types of dynamics, including detecting seizure activity in EEG [53]. We believe that using the wHVG as an objective, and the MOEA to fit mathematical models to data, could be a crucial tool to extend these analyses to better understand the mechanisms responsible for generating different dynamics.

The method we propose is different from probabilistic approaches such as DCM, Kalman filtering, or approximate Bayesian computation [21, 44, 45]. These methods optimise probability distributions for parameters given the data. As described in the introduction, these methods (as is the case for any approach) rely on certain simplifying assumptions, for example, Gaussianity in the case of DCM, and are not explicitly designed to search large parameter spaces. Though our method is designed for the latter, it is not a probabilistic framework, therefore the results here should also be carefully interpreted. The univariate parameter distributions we presented quantify where in parameter space non-dominated solutions were found, when starting from initial populations covering the whole space. Since the results are derived from multiple repeats of the EA, these distributions provide an approximation to the probability of finding parameters for which the model output closely approximates the data, given the algorithm used. The latter includes mechanisms such as mutation and crossover, which facilitate efficient exploration of parameter space. We propose it will be useful in the future to extend our approach into a probabilistic framework, for example, by incorporating uncertainties in the objective function evaluations into the calculation of dominance [54, 55]. Furthermore, we believe an additional potential use of the algorithm we present is to complement probabilistic schemes like DCM by using the output of the MOEA20 to formulate priors that are informative from a model dynamics perspective.

## Supporting information

**S1 Fig. Normalised convergence metrics of optimisation for different population sizes.**
(PDF)

**S2 Fig. PSD, wHVG and amplitude distributions derived from optimal model simulations for all control subjects, using each algorithm.**
(PDF)

**S3 Fig. Example of the SOEA45 missing the dominant alpha peak in favour of better fitting the 2–45Hz range.**
(PDF)

**S4 Fig. Comparison of normalised objective scores across the four algorithms.**
(PDF)

**S5 Fig. Distributions of amplitudes and Hurst exponents calculated from optimal model simulations.**
(PDF)

**S6 Fig. SOEA20 and SOEA45 parameters in 2-d space after a multi-dimensional scaling was applied to the parameters in the full space.**
(PDF)

**S7 Fig. SOEA20 and MOEA20 parameters in 2-d space after a multi-dimensional scaling was applied to the parameters in the full space.**
(PDF)

**S8 Fig. SOEA45 and MOEA20 parameters in 2-d space after a multi-dimensional scaling was applied to the parameters in the full space.**
(PDF)

**S9 Fig. MOEA20 and MOEA45 parameters in 2-d space after a multi-dimensional scaling was applied to the parameters in the full space.**
(PDF)

**S10 Fig. Parameter distributions recovered from resting EEG of all control subjects using the SOEA20 approach.**
(PDF)

**S11 Fig. Parameter distributions recovered from resting EEG of all control subjects using the SOEA45 approach.**
(PDF)

**S12 Fig. Parameter distributions recovered from resting EEG of all control subjects using the MOEA20 approach.**
(PDF)

**S13 Fig. Parameter distributions recovered from resting EEG of all control subjects using the MOEA45 approach.**
(PDF)

**S14 Fig. Parameter distributions recovered from simulated subjects.**
(PDF)

**S15 Fig. Distributions of all parameters recovered from FE and control EEG data.**
(PDF)

**S16 Fig. Comparison of mean membrane potentials and firing rates of excitatory and inhibitory populations at optimal parameters.**
(PDF)

**S17 Fig. Parameter distributions recovered from the SWD EEG data, using the MOEA20 approach.**
(PDF)

## Acknowledgments

We would like to thank Petroula Laiou, Alex Shaw and Luke Tait for valuable discussions relating to this work. We would like to dedicate this work to the memory of our dear friend and colleague Professor Ozgur E. Akman.

## Author Contributions

**Conceptualization:** Dominic M. Dunstan, Marc Goodfellow.

**Formal analysis:** Dominic M. Dunstan, Marc Goodfellow.

**Investigation:** Mark P. Richardson, Eugenio Abela.

**Methodology:** Dominic M. Dunstan, Ozgur E. Akman, Marc Goodfellow.

**Project administration:** Marc Goodfellow.

**Resources:** Marc Goodfellow.

**Software:** Dominic M. Dunstan, Marc Goodfellow.

**Supervision:** Ozgur E. Akman, Marc Goodfellow.

**Validation:** Dominic M. Dunstan, Marc Goodfellow.

**Visualization:** Marc Goodfellow.

**Writing – original draft:** Dominic M. Dunstan, Marc Goodfellow.

**Writing – review & editing:** Dominic M. Dunstan, Mark P. Richardson, Eugenio Abela, Ozgur E. Akman, Marc Goodfellow.

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
