## [Decision Letter · Decision Letter 0]

7 Dec 2022

Dear Mr Dunstan,

Thank you very much for submitting your manuscript "Global nonlinear approach for mapping parameters of neural mass models" for consideration at PLOS Computational Biology.

As with all papers reviewed by the journal, your manuscript was reviewed by members of the editorial board and by several independent reviewers. In light of the reviews (below this email), we would like to invite the resubmission of a significantly-revised version that takes into account the reviewers' comments.

We cannot make any decision about publication until we have seen the revised manuscript and your response to the reviewers' comments. Your revised manuscript is also likely to be sent to reviewers for further evaluation.

Sincerely,

Andrea E. Martin, Ph.D.

Academic Editor

PLOS Computational Biology

Marieke van Vugt

Section Editor

PLOS Computational Biology

Reviewer's Responses to Questions

**Comments to the Authors:**

Reviewer #1: See attached PDF

Reviewer #2: MS#: PCOMPBIOL-D-22-01564

Title: Global nonlinear approach for mapping parameters of neural mass models

Authors: Dunstan et al.

Abstract

The general context of this study is the identification of neural mass model (NMM) parameters from real data. Authors introduce a multi-objective optimization method based on evolutionary algorithms (MOEA) combined with the weighted horizontal visibility graph (wHVG) method. A major advantage is that it does not require simplifying procedures like the linearization of models which are formally nonlinear dynamical systems. They apply the methods to two datasets. First, they used eye-closed resting EEG recorded in 20 healthy subjects. Results indicate that it performs favorably compared to single objective approaches. Second, they analyze differences in the alpha rhythm observed in EEG recorded from 20 persons with epilepsy. Results suggest that the mean excitatory synaptic gain parameter is reduced in people with epilepsy compared to control. A conclusion is the MOEA method can be used to detect pathological rhythms in EEG recordings.

Major comments

This is an original and timely study in a context of computation models of brain activity have gained increasing interest in the field of neurosciences and neurological disorders. The paper is clear, well written and results are presented rigorously. Authors should address the following points to enhance their manuscript.

I think, before discussing how to choose the model parameters, it is important to discuss how to choose the model. Because, it is the model structure that determines the global dynamical space, not the parameters themselves. Already, the simple neural models with one excitatory and one inhibitory neuron populations (like the Liley model here) ignore the existence of different types of inhibitory interneurons in neocortex (see for instance Bensaid et al. (2019) COALIA: A Computational Model of Human EEG for Consciousness Research. Front. Syst. Neurosci. 13:59. doi: 10.3389/fnsys.2019.00059).

My understanding is that the model output is the mean membrane potential of the subpopulation of excitatory neurons he(t). This contrasts with many other studies in which the model output is taken to be the summated PSPs, both excitatory and inhibitory, at the level of main cells (typically pyramidal cells in cortical regions). Authors should explain their choice. Additionally, it would be informative to see simulated time series after parameter identification, compared with real signals.

In line with the previous comment, NMM output is a proxy of the local field potential generated in a population of neurons (i.e. a source) comprising sub-populations of excitatory and inhibitory neurons. Authors compare the model output to scalp EEG data recorded from 2 electrodes (occipital for healthy subjects and frontal for people with epilepsy). They should better argue why such a direct comparison is valid in a context where scalp EEG signals reflect the activity of all sources in the brain depending on the source-electrode distance and the source orientation (in accordance with the EEG forward solution).

Regarding the results section, I would suggest to test SOEA and MOEA methods on signals simulated by the model for different parameter configurations producing alpha activity. This would provide a ground-truth about parameters to recover and thus a complementary test of the performance of methods.

In the discussion, it would be nice to address the following point: can the multi-objective optimization algorithm introduced in the article be used to estimate parameters of a NMM from non-stationary data like sporadic epileptic spikes or evoked potentials, for instance? I imagine that wHVG feature can still be used. How would the other objectives be constructed?

Before interpreting the observations on the average synaptic gain (line 471-487), it would be interesting to a bit more clinical information about the 20 people with epilepsy. It is well known that the key variable in the NMMs is the ratio between excitation and inhibition (ePSP and iPSP). Is the decrease of excitation identified from EEG associated with a value of inhibition similar to that identified in HCs. In other words, is the E/I ratio decreased in EEGs recorded in people with epilepsy? In addition, it is mentioned that one of the epilepsy patients is diagnosed with an absence epilepsy. What about the others? Is the cohort homogenous or heterogenous in term of aetiology? These aspects are important for the understanding and interpretation of the excitation decrease.

Minor comments

Abstract: In the last paragraph of the abstract, it reads that different EEG states recoded in healthy controls (resting EEG) and in epileptic patients (alpha rhythm EEG). The author summary say it is resting, into says it is resting. Then section 2.2 talks about alpha… The study compares …. Please say that it is “alpha activity recoded during the eye closed resting state”

Figure 1: numbering of the objectives. Inconsistency with the text (pg8)

Figure 2: firstly, why are the base lines of normalized power diagrams different? Secondly, why are the y-axes labels in 2d, 2e, 2i, and 2j “normalized frequency”? Finally, visually it is hard to accept that wHVG criteria improved the fitting. Figure 3 zooms into the data presented in 2a, but I find the match is not very convincing. Can authors propose an improvement in the model or algorithm or improve the match?

Pg3, line 60: Clarify the sentence “We might therefore expect neural mass postsynaptic potentials (PSPs) to be parameterised using values that are different from the values used to parameterise neuronal PSPs.”

Figure 3: why are the y-axes labels 3b “normalized frequency”?

Face value or silhouette value? Both terms are used.

The “Results” section could be organized in subsections. It would be easier to follow.

**Have the authors made all data and (if applicable) computational code underlying the findings in their manuscript fully available?**

Reviewer #1: **No: **Authors state that they will make the code available on acceptance but is not currently available

Reviewer #2: Yes

PLOS authors have the option to publish the peer review history of their article (what does this mean?). If published, this will include your full peer review and any attached files.

Reviewer #1: No

Reviewer #2: No
---

## [Decision Letter · Decision Letter 1]

1 Mar 2023

Dear Mr Dunstan,

We are pleased to inform you that your manuscript 'Global nonlinear approach for mapping parameters of neural mass models' has been provisionally accepted for publication in PLOS Computational Biology.

Best regards,

Andrea E. Martin, Ph.D.

Academic Editor

PLOS Computational Biology

Marieke van Vugt

Section Editor

PLOS Computational Biology

Reviewer's Responses to Questions

**Comments to the Authors:**

Reviewer #1: I am satisfied that the authors have addressed all of my concerns

Reviewer #2: The authors have adequately addressed the comments we made on the first version of the manuscript. They have also taken our suggestions into account.

**Have the authors made all data and (if applicable) computational code underlying the findings in their manuscript fully available?**

Reviewer #1: Yes

Reviewer #2: Yes

PLOS authors have the option to publish the peer review history of their article (what does this mean?). If published, this will include your full peer review and any attached files.

Reviewer #1: No

Reviewer #2: No

---

## [Editor Report · Acceptance letter]

20 Mar 2023

PCOMPBIOL-D-22-01564R1 

Global nonlinear approach for mapping parameters of neural mass models

Dear Dr Dunstan,

I am pleased to inform you that your manuscript has been formally accepted for publication in PLOS Computational Biology. Your manuscript is now with our production department and you will be notified of the publication date in due course.

With kind regards,

Dorothy Lannert
